# PAG: Multi-Turn Reinforced LLM Self-Correction with Policy as Generative Verifier

## Abstract

Large Language Models (LLMs) have demonstrated impressive capabilities in complex reasoning tasks, yet they still struggle to reliably verify the correctness of their own outputs. Existing methods to this verification challenge often depend on separate verifier models or require multi-stage self-correction training pipelines, which limit scalability. In this paper, we propose Policy as Generative Verifier (PAG), a simple and effective framework that empowers LLMs to self-correct by alternating between policy and verifier roles within a unified multi-turn reinforcement learning (RL) paradigm. Distinct from prior approaches that always generate a second solution regardless of model confidence, PAG introduces a selective revision mechanism: the model revises its answer only when its own generative verification step detects an error. This verify-then-revise workflow not only alleviates model collapse but also jointly enhances both reasoning and verification abilities. Extensive experiments on mathematical reasoning benchmarks highlight PAG's dual advancements: as a policy, it enhances direct generation and self-correction accuracy; as a verifier, its self-verification outperforms majority voting.

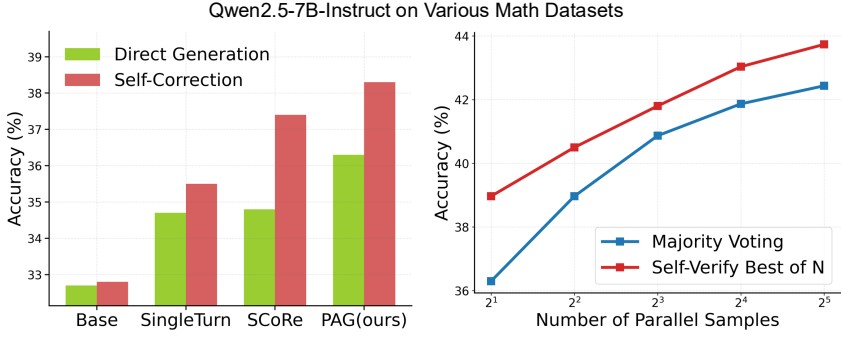

Figure 1: **Performance highlight of PAG.**

## 1 Introduction

Large Language Models (LLMs) have demonstrated remarkable proficiency in solving complex reasoning problems (Cobbe et al., 2021; Zelikman et al., 2022; OpenAI, 2024; Guo et al., 2025; Seed et al., 2025; Comanici et al., 2025; Kimi et al., 2025). Despite these advances, a significant challenge remains: although LLMs are capable of generating complex solutions, they often struggle to reliably verify the correctness of their own reasoning (Huang et al., 2023; Tyen et al., 2024). This verification gap presents a critical obstacle in domains where solution accuracy is essential, such as advanced mathematical reasoning (Hendrycks et al., 2021; Lewkowycz et al., 2022; Phan et al., 2025), scientific research (Luo et al., 2025), and dependable decision support systems (Li et al., 2024; Shool et al., 2025).

To bridge this verification gap, previous research has primarily explored two main directions. The first centers on the use of standalone verifier models, which are explicitly trained to evaluate the quality of generated solutions (Cobbe et al., 2021; Lightman et al., 2023; Zhang et al., 2025b;a). These verifiers typically rely on human-labeled data or leverage more advanced "LLM-as-a-judge" systems for supervision (Wu et al., 2024). While this strategy can be effective, it requires deploying an additional

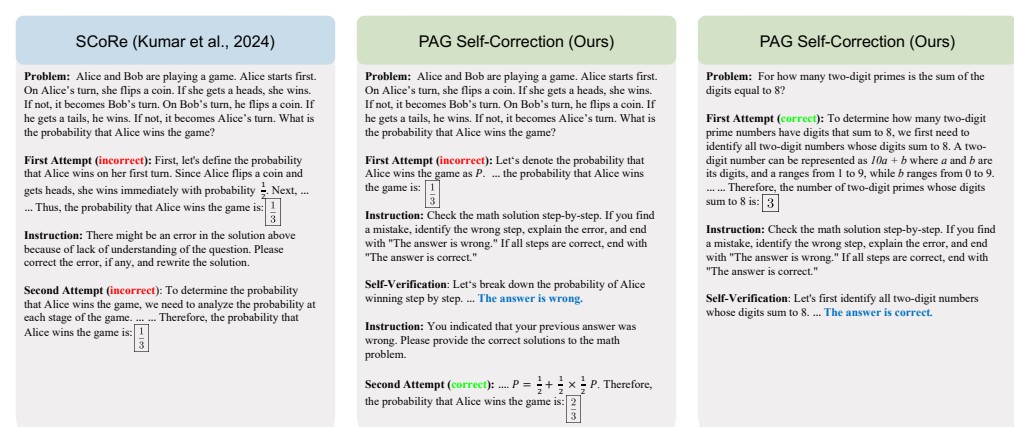

Figure 2: **Examples of PAG self-correction.**

model, thereby increasing both computational costs and system complexity. The second line of work seeks to enhance the self-correction abilities of LLMs themselves. Early studies adopted supervised fine-tuning (SFT) on datasets containing errors and their corresponding corrections (Zelikman et al., 2022; Madaan et al., 2023; Chen et al., 2023). However, SFT-based methods often suffer from distribution shift (Kumar et al., 2024): models trained on static datasets may fail to generalize to new types of errors that arise as the model evolves. As a result, reinforcement learning (RL) has gained traction as a more flexible framework (Kumar et al., 2024; Yuan et al., 2025; Ma et al., 2025), allowing models to learn directly from their own, dynamically generated responses.

Nevertheless, effectively leveraging reinforcement learning (RL) for self-correction introduces several challenges. One notable issue is the phenomenon of "model collapse" (Kumar et al., 2024), where the LLM, when prompted to improve its output, may only make minor or superficial changes, or sometimes simply repeat its initial response. This can occur if the RL objective does not sufficiently encourage substantive improvements, or if the model has difficulty distinguishing between genuine corrections and paraphrasing, which can hinder learning progress. To address this, Kumar et al. (2024) has explored multi-stage RL training approaches. These approaches often rely on carefully designed learning curricula, which can increase the complexity of the training process.

We argue that such multi-stage approaches (Kumar et al., 2024) may not be necessary. This paper introduces **Policy as Generative Verifier (PAG)**, a simple yet effective method grounded in a key insight: rather than always generating a second solution, it is more effective to employ a *selective revision mechanism*—the model only revises its answer when its own self-verification explicitly identifies an error.

In PAG, the model first acts on the role of a *generative verifier*, critically evaluating its own initial answer. Only when this self-verification step identifies an error does the model proceed to generate a revised response. As illustrated in Figure 2, this "verify-then-revise" workflow fundamentally distinguishes PAG from previous self-correction methods that always perform a second attempt regardless of the model's confidence. This design offers two main advantages. First, by explicitly introducing a verification step, PAG effectively mitigates model collapse, encouraging genuine self-reflection rather than superficial rephrasing of the initial answer. Second, the integrated alternation between generation and verification roles enables the model to jointly improve both its reasoning and verification abilities within a unified training framework.

Our main contributions are summarized as follows:

- We propose **Policy as Generative Verifier (PAG)**, a novel and streamlined framework that enables large language models (LLMs) to achieve self-correction by alternately switching between policy and generative verifier roles.

- We demonstrate that PAG can be efficiently implemented using multi-turn reinforcement learning, eliminating the need for complex multi-stage training or warm-up phases.

- Our experiments on mathematical reasoning demonstrate that PAG, by unifying policy and verifier capabilities into a single model, enhances self-correction performance while enabling a self-verification best-of-N strategy that outperforms majority voting.

## 2 RELATED WORK

**LLM Self-Correction.**  Recent advances (Liu et al., 2023; Jain et al., 2024; Chen et al., 2023) have demonstrated that large language models (LLMs) exhibit self-improvement capabilities when provided with external feedback in agentic tasks or code repair tasks. In this work, we focus on intrinsic self-correction, where the model improves its responses during inference without access to any external feedback. The underlying motivation is that LLMs often possess latent knowledge that is difficult to elicit through standard next-token generation, yet self-correction can distill this hidden knowledge into improved outputs (Yuan et al., 2025; Song et al., 2024; Huang et al., 2025). Prior studies (Shinn et al., 2023; Madaan et al., 2023) have shown that prompting can sometimes elicit self-correction abilities in LLMs; however, such intrinsic self-correction is neither robust nor significant in complex domains (Huang et al., 2023; Tyen et al., 2024). Several approaches (Saunders et al., 2022; Ye et al., 2023; Qu et al., 2024) have attempted to leverage supervised fine-tuning (SFT) by distilling reranked or filtered generations from human or stronger model generated data, but the resulting self-correction abilities suffer from distribution shift and limited generalization (Kumar et al., 2024). Complementary to these, ReVISE (Lee et al., 2025) introduces intrinsic self-verification: uses a two-stage preference-learning curriculum—first learning to verify initial reasoning by choosing between stop or refine, then learning to correct when verification predicts error by preferring refined trajectories—thus internalizing when-to-stop and how-to-refine and yielding consistent gains on math and code benchmarks.

Reinforcement learning (RL) has emerged as a powerful solution to the limited generalization of supervised fine-tuning (SFT). SCoRe (Kumar et al., 2024) pioneers multi-turn RL for self-correction, but discovers that naively applying multi-turn RL leads to model collapse, where the model's second attempt exhibits minimal deviation from the first. To address this, they propose a two-stage training procedure. The first stage is a warm-up phase that uses RL to decouple the model's behavior across the two attempts by optimizing for second-attempt accuracy while constraining the first attempt to match the base model's distribution. The second stage then applies multi-turn RL training. Recent concurrent works (Xiong et al., 2025; Ma et al., 2025) propose self-rewarding RL frameworks similar to our approach, but they require extensive supervised fine-tuning and only explore single-turn RL. In contrast, our approach jointly trains the policy and verifier via multi-turn RL, and, crucially, does not require any warm-up phase—training can begin directly from existing instruction-tuned models.

**Generative Verifier.**  Traditional verifiers are predominantly discriminative, typically trained as classifiers to distinguish between prefer and disprefer responses (Cobbe et al., 2021; Lightman et al., 2023; Zhang et al., 2025b; Sun et al., 2025). In contrast, generative verifiers (Zhang et al., 2025a; Mahan et al., 2024) leverage the text generation capabilities of large language models (LLMs), adopting a next-token prediction paradigm for verification. This approach enables the verifier to perform chain-of-thought reasoning, thereby facilitating more nuanced and effective verification processes. Recent works (Xu et al., 2025; Liu et al., 2025; Chen et al., 2025; Shi & Jin, 2025) have begun to explore reinforcement learning (RL) for training generative verifiers. However, most prior efforts focus exclusively on either policy optimization or verifier training in isolation. In this work, we propose a unified RL framework that simultaneously optimizes both the the policy and the generative verifier and enabling them to co-evolve and reinforce each other's capabilities throughout the training process.

**Multi-Turn RL.**  Recent studies have investigated multi-turn reinforcement learning (RL) for large language models (LLMs) from various perspectives, including preference-based (Shani et al., 2024; Xiong et al., 2024; Zhou et al., 2025), value-based (Zhou et al., 2024; Abdulhai et al., 2023), and on-policy (Wang et al., 2025; Kumar et al., 2024) approaches. Multi-turn RL is emerging as a highly promising direction for developing LLM agents (Qin et al., 2025), though the field remains in its early stages.

## 3 POLICY AS GENERATIVE VERIFIER

This section introduces the Policy as Generative Verifier (PAG) framework, which enables large language models (LLMs) to perform multi-turn self-correction. As illustrated in Figure 3, PAG alternates the LLM between two distinct roles: a *policy* that proposes candidate solutions, and a *verifier* that critically evaluates the generated solutions through self-verification. This iterative process continues until the solution is verified as correct or a maximum number of turns is reached. We begin by formally outlining the framework and problem formulation.

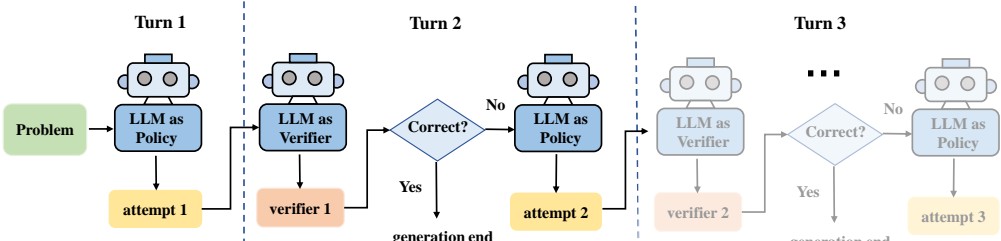

Figure 3: **Overview of the Policy as Generative Verifier (PAG) framework.** The LLM alternates between a policy role (generating solutions) and a generative verifier role (evaluating its own solutions) in a multi-turn process. This iterative refinement continues until self-verification is correct or a maximum number of turns is reached.

## 3.1 Framework and Problem Formulation

Given an input problem $\mathbf{x}$, the LLM policy $\pi_\theta$ first generates an initial solution attempt $\mathbf{y}_1$. In each subsequent turn $t$, the model produces a self-verification $\mathbf{v}_t$ for the previous attempt $\mathbf{y}_{t-1}$. if the verifier $\mathbf{v}_t$ deems attempt $\mathbf{y}_{t-1}$ correct, the process terminates. Otherwise, the LLM generates a revised attempt $\mathbf{y}_t$, and the cycle repeats. The process continues until either the attempt is verified as correct or a predefined maximum number of turns $T_{\max}$ is reached. During training, both solution attempt and verification receive reward signals from a *external ground-truth verifier*. At inference time, the model must rely on itself, without any external feedback.

Formally, for each input $\mathbf{x}$, the model generates a trajectory $\tau_\mathbf{x} = (\mathbf{x}, \mathbf{y}_1, \mathbf{v}_2, \mathbf{y}_2, \mathbf{v}_3, \ldots)$ until termination. Let $N_A(\tau_\mathbf{x})$ and $N_V(\tau_\mathbf{x})$ denote the number of solution attempts and verifications in the trajectory, respectively. We define $R_y(\mathbf{y}_i, \mathbf{x})$ as the reward for the $i$-th attempt and $R_v(\mathbf{v}_j, \mathbf{y}_{j-1}, \mathbf{x})$ as the reward for the $j$-th verification. Both rewards are binary: $R_y(\mathbf{y}_i, \mathbf{x}) = 1$ if the attempt is correct and 0 otherwise, while $R_v(\mathbf{v}_j, \mathbf{y}_{j-1}, \mathbf{x}) = 1$ if the verification accurately identifies the correctness of the attempt and 0 otherwise. The overall training objective is formulated as:

$$\max_\theta \ \mathbb{E}_\mathbf{x}\left[\mathbb{E}_{\tau_\mathbf{x} \sim \pi_\theta}\left[\sum_{i=1}^{N_A(\tau_\mathbf{x})} R_y(\mathbf{y}_i, \mathbf{x}) + \sum_{j=1}^{N_V(\tau_\mathbf{x})} R_v(\mathbf{v}_{j+1}, \mathbf{y}_j, \mathbf{x})\right]\right] \tag{1}$$

This objective encourages the model to generate both correct solutions and self-verifications, facilitating iterative refinement and early termination upon correctness.

A key innovation of our framework is a *selective revision mechanism* that uses the model itself as a generative verifier. Unlike approaches like SCoRe (Kumar et al., 2024) that always attempt revision, PAG revises an output only when its self-verification deems it wrong. This selectivity is crucial in multi-turn reinforcement learning. It prevents the model collapse seen in SCoRe, where unconditional revisions can cause the model to merely generate superficial variations of its initial response rather than correcting it. By conditioning revision on self-verification, PAG improves self-correction and eliminates SCoRe's two-stage RL procedure, resulting in a more streamlined and efficient training process.

## 3.2 Multi-Turn Reinforcement Learning

Standard single-turn reinforcement learning (RL) algorithms can be seamlessly extended to multi-turn LLM generation by treating the entire sequence of turns as a single, concatenated trajectory. In this formulation, the model's outputs from each turn are concatenated together, with rewards assigned at the end of each corresponding segment. This design enables us to leverage existing PPO (Schulman et al., 2017) algorithms without requiring architectural modifications, providing a straightforward and effective approach for extending RL to multi-turn scenarios.

We use the PPO algorithm to optimize the objective in Equation 1. For a batch of $N$ trajectories, the PPO objective is:

$$\mathcal{J}_{\text{PPO}}(\theta) = \frac{1}{N}\sum_{i=1}^{N}\frac{1}{|\tau_i|}\sum_{t=1}^{|\tau_i|}\left[\min\left(r_{i,t}(\theta)\hat{A}_{i,t}, \text{clip}(r_{i,t}(\theta), 1-\epsilon, 1+\epsilon)\hat{A}_{i,t}\right)\right] \tag{2}$$

where $r_{i,t}(\theta)$ is the importance sampling ratio between the current and old policies, $\epsilon$ is the clipping hyperparameter, and $\hat{A}_{i,t}$ is the advantage estimate for token $\tau_{i,t}$, typically computed using GAE (Schulman et al., 2015). In practice, as suggested by prior work (Xu et al., 2024; Andrychowicz et al., 2021; Raffin et al., 2021), these advantages are normalized across the batch to stabilize training: $\hat{A}_{i,t}^{\mathrm{norm}} = (\hat{A}_{i,t} - \mu)/\sigma$, where $\mu$ and $\sigma$ are the mean and standard deviation of the advantages within the batch, respectively.

The PAG framework, however, presents a unique challenge: the LLM alternates between two roles—policy and verifier—with fundamentally different objectives. To address this, we introduce three key adaptations to the standard PPO algorithm.

First, we employ a **turn-independent optimization** strategy. In this approach, advantages are not propagated from later turns to earlier ones; instead, optimization is confined to the current turn. This is equivalent to setting the inter-turn discount factor to zero during GAE computation. We empirically find this design to be crucial for achieving stable and effective training in the PAG framework.

Second, inspired by (Kumar et al., 2024), we introduce a **bonus reward** during policy turns to encourage self-correction. The bonus for the current attempt, $\mathbf{y}_k$, is proportional to the performance improvement over the previous attempt, $\mathbf{y}_{k-1}$:

$$R_y(\mathbf{y}_k, \mathbf{x}) \leftarrow R_y(\mathbf{y}_k, \mathbf{x}) + b(\mathbf{y}_k), \quad \text{where} \quad b(\mathbf{y}_k) = \alpha \cdot \big(R_y(\mathbf{y}_k, \mathbf{x}) - R_y(\mathbf{y}_{k-1}, \mathbf{x})\big). \quad (3)$$

where $\alpha > 0$ is a scaling factor. This bonus is added to the reward of the policy generation turn.

Third, we propose **RoleAdvNorm**, an advantage normalization technique tailored for our multi-role setting. Whereas standard PPO normalizes advantages over the entire trajectory, RoleAdvNorm normalizes them separately for the policy and verifier roles:

$$\hat{A}_{i,t,\mathrm{role}}^{\mathrm{norm}} = \frac{\hat{A}_{i,t} - \mu_{\mathrm{role}}}{\sigma_{\mathrm{role}}}, \quad \text{role} \in \{\text{policy, verifier}\} \quad (4)$$

where $\mu_{\mathrm{role}}$ and $\sigma_{\mathrm{role}}$ are the mean and standard deviation of advantages associated with a given role within the batch. This role-specific normalization prevents the gradients from different roles from interfering with each other, leading to more stable updates. Our experiments show that RoleAdvNorm, especially when combined with the reward bonus, enhances the model's self-correction capabilities.

## 4 EXPERIMENTS

We mainly focus on mathematical reasoning tasks. We also conduct experiments on logical and coding reasoning tasks, with the results detailed in the Section 4.5 and 4.6. All methods are trained using PPO with identical training hyperparameters. The training details is detailed in Appendix C. Unless otherwise specified, all our experiments are conducted with two training turns and two evaluation turns.

**Base Models.** We adopt Qwen2.5-1.5B-Instruct, Qwen2.5-7B-Instruct (Yang et al., 2024a), and Llama3-8B-Instruct (Grattafiori et al., 2024) as our base models. We also include an experiment on Qwen2.5-32B-Instruct in Appendix B.1. Unlike prior works (Kumar et al., 2024; Xiong et al., 2025), we *do not* employ any additional SFT finetuning or warm-up training.

**Training and Evaluation Data.** For base models with relatively limited capabilities, such as Qwen2.5-1.5B-Instruct and Llama3-8B-Instruct, we use the MATH (Hendrycks et al., 2021) training set, which contains 7.5K samples. For more capable models, such as Qwen2.5-7B-Instruct, we employ DAPO17K (Yu et al., 2025) for training. We evaluate on a broader set of mathematical benchmarks including MATH500, MinevaMATH (Lewkowycz et al., 2022), AIME2024, and AIME2025.

**Baselines.** Our baselines include: (1) **SingleTurn**, model only generate once during the RL training; (2) **Direct MultiTurn**, model directly generate two turns without self-verification; (3) **SCoRe** (Kumar et al., 2024), builds on Direct Multiturn, but with two-stage RL training for better self-correction performance; We re-implemented SCoRe in our codebase, as the official implementation is not publicly available.

### 4.1 POLICY RESULTS

We first present the results for the model acting as a policy. We evaluate policy performance using two primary metrics: **Acc.@t1** (direct generation accuracy) and **Acc.@final** (self-correction accuracy).

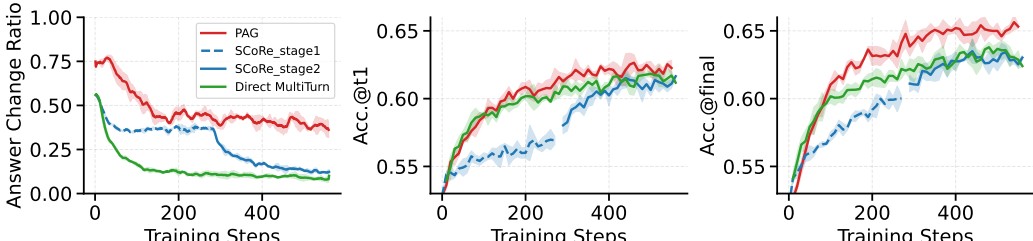

Figure 4: **Training dynamics of PAG on Qwen2.5-1.5B-Instruct (3 seeds). Left:** *Answer change ratio* quantifies model collapse as the proportion of responses where the second-turn answer differs from the first. Direct MultiTurn rapidly declines, indicating severe collapse. SCoRe partially alleviates this through two-stage RL, while PAG's selective revision mechanism effectively prevents collapse and achieves higher Acc.@t1 and Acc.@final; **Middle:** Acc.@t1; **Right:** Acc.@final.

Table 1: **Self-Correction Performance (%) of PAG on Qwen2.5-1.5B-Instruct.** (subscripted with ± indicating standard deviation over 3 seeds.) PAG improves the revision efficiency.

| Method | Acc.@t1 | Acc.@final | $\Delta$(t1,final) | $\Delta^{i\to c}$ | $\Delta^{c\to i}$ | $\hat{\Delta}^{i\to c}$ | $\hat{\Delta}^{c\to i}$ |
|---|---|---|---|---|---|---|---|
| Base Model | 52.9 | 52.0 | -0.9 | 2.9 | 3.8 | 2.9 | 3.8 |
| Direct MultiTurn | $61.7_{\pm0.2}$ | $63.8_{\pm0.4}$ | $2.1_{\pm0.3}$ | $3.6_{\pm0.4}$ | $1.5_{\pm0.3}$ | $3.6_{\pm0.4}$ | $1.5_{\pm0.3}$ |
| SCoRe | $61.5_{\pm0.7}$ | $63.5_{\pm0.9}$ | $2.0_{\pm0.5}$ | $4.4_{\pm0.3}$ | $2.4_{\pm0.5}$ | $4.4_{\pm0.3}$ | $2.4_{\pm0.5}$ |
| PAG | $\mathbf{62.5}_{\pm0.2}$ | $\mathbf{65.6}_{\pm0.5}$ | $\mathbf{3.1}_{\pm0.4}$ | $\mathbf{4.5}_{\pm0.6}$ | $\mathbf{1.4}_{\pm0.3}$ | $\mathbf{14.9}_{\pm0.7}$ | $\mathbf{4.6}_{\pm0.6}$ |

For PAG, Acc.@final is measured on a revised response only when the initial attempt is self-identified as incorrect; otherwise, Acc.@final equals Acc.@t1. For baselines like Direct MultiTurn and SCoRe, which always generate a second solution attemp, Acc.@final is the second-turn accuracy.

To analyze self-correction in detail, we use several diagnostic metrics. $\Delta$(t1,final) is the accuracy gain after correction. $\Delta^{i\to c}$ and $\Delta^{c\to i}$ are the fractions of all samples that flip from incorrect-to-correct and correct-to-incorrect, respectively. The hatted versions, $\hat{\Delta}^{i\to c}$ and $\hat{\Delta}^{c\to i}$, are normalized by the number of revised samples. Note that baselines revise all samples, while PAG only revises those that self-verification identifies as incorrect.

**PAG effectively mitigates model collapse by enhancing revision efficiency.** We measure the answer change ratio, which is the proportion of final answers (often a numerical value) that differ after a revision turn. Figure 4 plots the results of answer change ratio and Acc.@t1 and Acc.@final on MATH500, averaged over three runs with different random seeds (shaded areas denote standard deviation). As shown, baselines like Direct MultiTurn suffer from rapid collapse as the answer change ratio declines sharply. PAG's selective revision mechanism—revising only when an answer is deemed incorrect—fundamentally addresses this. PAG also makes revision more efficiency. As shown in Table 1, PAG boosts the incorrect-to-correct revision rate ($\hat{\Delta}^{i\to c}$) to 14.9%, far surpassing SCoRe's 4.4%, demonstrating that it learns to correct errors effectively, not just repeat them. In contrast, the base model fails to achieve meaningful gains from a second reasoning turn, underscoring that self-correction is a non-trivial capability that must be explicitly trained. By preventing collapse and improving revision efficiency, PAG yields superior Acc.@final performance.

**PAG achieves superior self-correction performance across different LLMs.** As shown in Table 2, PAG demonstrates robust self-correction performance, consistently achieving the highest Acc.@final across various models. It delivers strong gains on both Qwen2.5-1.5B and Llama3-8B, showcasing its effectiveness across different model architectures. With Qwen2.5-7B, PAG achieves state-of-the-art results on mathematical benchmarks, attaining an Acc.@final of 82.3% on MATH500 and the best overall average performance (38.3%). These results confirm that PAG provides robust improvements to LLM self-correction.

## 4.2 GENERATIVE VERIFIER RESULTS

Table 2: **Performance of PAG on three Models. PAG consistently leads in Acc.@final.**, evaluated using Avg@32. For each benchmark, results are highlighted with underline and **bold** for the best Acc.@t1 and Acc.@final, respectively. Subscripts on Acc.@final denote the gain relative to Acc.@t1.

| Model | Method | Metric | MATH500 | MinervaMath | AIME24 | AIME25 | Average |
|---|---|---|---|---|---|---|---|
| Qwen2.5-1.5B-Instruct | Base | Acc.@t1 | 53.2 | 17.4 | 3.0 | 1.3 | 18.7 |
| | | Acc.@final | $52.4_{-0.8}$ | $16.6_{-0.8}$ | $2.8_{-0.2}$ | $1.4_{+0.1}$ | $18.3_{-0.4}$ |
| | Single-Turn | Acc.@t1 | 61.6 | 21.9 | 4.1 | 1.6 | 22.3 |
| | | Acc.@final | $62.4_{+0.8}$ | $21.0_{-0.9}$ | $4.2_{+0.1}$ | $1.9_{+0.3}$ | $22.4_{+0.1}$ |
| | Direct Multi-Turn | Acc.@t1 | 62.1 | 21.3 | 2.8 | 1.6 | 22.0 |
| | | Acc.@final | $64.0_{+1.9}$ | $21.2_{-0.1}$ | $3.2_{+0.4}$ | $1.0_{-0.6}$ | $22.4_{+0.4}$ |
| | SCoRe | Acc.@t1 | 61.9 | 21.7 | 4.1 | 2.4 | 22.5 |
| | | Acc.@final | $63.9_{+2.0}$ | $20.9_{-0.8}$ | $\mathbf{5.7}_{+1.6}$ | $2.5_{+0.1}$ | $23.3_{+0.8}$ |
| | **PAG**(Ours) | Acc.@t1 | 62.5 | 21.6 | 5.0 | 2.3 | 22.9 |
| | | Acc.@final | $\mathbf{65.6}_{+3.1}$ | $\mathbf{23.0}_{+1.4}$ | $5.0_{+0.0}$ | $\mathbf{3.8}_{+1.5}$ | $\mathbf{24.4}_{+1.5}$ |
| Llama3-8B-Instruct | Base | Acc.@t1 | 28.7 | 13.9 | 0.3 | 0 | 10.7 |
| | | Acc.@final | $26.6_{-2.1}$ | $12.7_{-1.2}$ | $0.6_{+0.3}$ | $0.2_{+0.2}$ | $10.0_{-0.7}$ |
| | Single-Turn | Acc.@t1 | 32.4 | 16.2 | 0.5 | 0.1 | 12.3 |
| | | Acc.@final | $32.4_{+0.0}$ | $15.1_{-1.1}$ | $0.7_{+0.2}$ | $0.1_{+0.0}$ | $12.1_{-0.2}$ |
| | Direct Multi-Turn | Acc.@t1 | 33.3 | 18.0 | 2.1 | 0.1 | 13.4 |
| | | Acc.@final | $35.4_{+2.1}$ | $18.8_{+0.8}$ | $\mathbf{1.4}_{-0.7}$ | $0.2_{+0.1}$ | $14.0_{+0.6}$ |
| | SCoRe | Acc.@t1 | 32.0 | 18.0 | 1.0 | 0.1 | 12.8 |
| | | Acc.@final | $35.2_{+3.2}$ | $19.1_{+1.1}$ | $0.9_{-0.1}$ | $0.1_{+0.0}$ | $13.8_{+1.0}$ |
| | **PAG**(Ours) | Acc.@t1 | 35.1 | 18.8 | 0.5 | 0.2 | 13.7 |
| | | Acc.@final | $\mathbf{36.7}_{+1.6}$ | $\mathbf{20.6}_{+1.8}$ | $0.7_{+0.2}$ | $\mathbf{0.3}_{+0.1}$ | $\mathbf{14.6}_{+0.9}$ |
| Qwen2.5-7B-Instruct | Base | Acc.@t1 | 76.0 | 35.9 | 11.1 | 7.6 | 32.7 |
| | | Acc.@final | $76.1_{+0.1}$ | $35.6_{-0.3}$ | $11.8_{+0.7}$ | $7.8_{+0.2}$ | $32.8_{+0.1}$ |
| | Single-Turn | Acc.@t1 | 79.5 | 36.5 | 13.2 | 9.6 | 34.7 |
| | | Acc.@final | $80.2_{+0.7}$ | $37.1_{+0.6}$ | $14.1_{+0.9}$ | $10.6_{+1.0}$ | $35.5_{+0.8}$ |
| | Direct Multi-Turn | Acc.@t1 | 80.0 | 35.3 | 15.8 | 11.9 | 35.8 |
| | | Acc.@final | $81.2_{+1.2}$ | $35.4_{+0.1}$ | $17.3_{+1.5}$ | $\underline{13.8}_{+1.9}$ | $36.9_{+1.1}$ |
| | SCoRe | Acc.@t1 | 79.5 | 36.7 | 13.1 | 10.0 | 34.8 |
| | | Acc.@final | $81.5_{+2.0}$ | $\mathbf{37.7}_{+1.0}$ | $16.6_{+3.5}$ | $14.0_{+4.0}$ | $37.4_{+2.6}$ |
| | **PAG**(Ours) | Acc.@t1 | 80.3 | 37.0 | 16.6 | 11.4 | 36.3 |
| | | Acc.@final | $\mathbf{82.3}_{+2.0}$ | $37.2_{+0.2}$ | $\mathbf{18.4}_{+1.8}$ | $\mathbf{15.1}_{+3.7}$ | $\mathbf{38.3}_{+2.0}$ |

We evaluate the performance of our model as a generative verifier. For generative verifier evaluation, we report three key metrics: **CR** (Correct Recall, the proportion of correct answers that are correctly identified by the verifier), **WR** (Wrong Recall, the proportion of incorrect answers that are correctly identified as wrong), and **Acc** (Verifer Accuracy, the overall accuracy of the verifier across all samples). Additionally, we report results for **self-verify BoN**, where the model generates $N$ candidate responses in parallel for each prompt and subsequently acts as a verifier to assess the correctness of each response. The final answer is selected as the response that receives the highest probability for the "The answer is correct."

**PAG achieves high verifier performance with both self-generated and offline datasets.** Table 3 and Table 4 present the performance of the generative verifier on both self-generated responses from the MATH500 dataset and the offline RewardBench mathprm benchmark, respectively. The results demonstrate that baseline models (Qwen2.5-1.5B/7B-Instruct) exhibit limited self-verification capabilities. Notably, the SCoRe baseline, which also involves answer revision, shows even poorer verifier performance than the base model for the 1.5B model as shown in Table 3. While SCoRe provides some improvement for the 7B model, its gains are modest compared to those achieved by PAG. This suggests that merely allowing answer revision, as SCoRe does, is not a consistently effective or effective strategy for developing strong self-verification capabilities. However, after incorporating the PAG framework, both models achieve substantial improvements across all metrics. Specifically, Verifier Accuracy on MATH500 increases by over 50% for the 1.5B model and

Table 3: **Performance of Self-verification.**

| Method | Acc. | C.R. | W.R. |
|---|---|---|---|
| Qwen2.5-1.5B-Instruct | 30.3 | 52.9 | 4.4 |
| + SCoRe | 8.2 | 14.3 | 0.4 |
| **+ PAG** | **81.7** | **91.2** | **65.8** |
| Qwen2.5-7B-Instruct | 47.5 | 57.8 | 14.3 |
| + SCoRe | 62.5 | 74.6 | 15.2 |
| **+ PAG** | **90.7** | **94.2** | **76.4** |

Table 4: **Performance on RewardBench.** Scores* are taken from RewardBench report.

| Model | Score |
|---|---|
| Qwen2.5-1.5B-Instruct | 23.9 |
| Qwen2.5-7B-Instruct | 37.8 |
| gemini-1.5-pro-0924 | 83.9* |
| Meta-Llama-3.1-70B-Instruct | 76.4* |
| gpt-4-0125-preview | 76.3* |
| **Qwen2.5-1.5B-Instruct + PAG** | 78.5 |
| **Qwen2.5-7B-Instruct + PAG** | 86.6 |

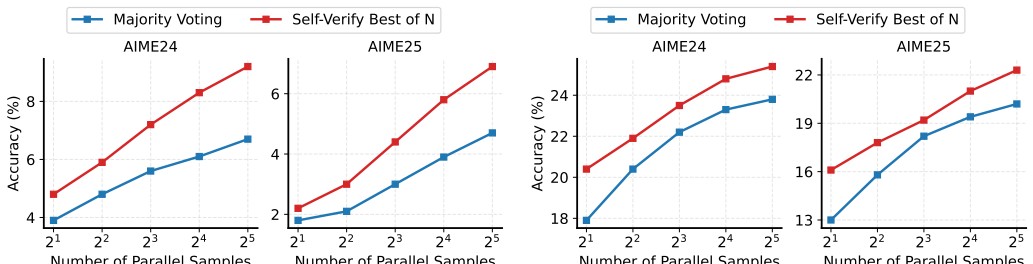

Figure 5: **PAG self-verify BoN outperforms majority voting. Left**: Qwen2.5-1.5B-Instruct + **PAG**; **Right**: Qwen2.5-7B-Instruct + **PAG**.

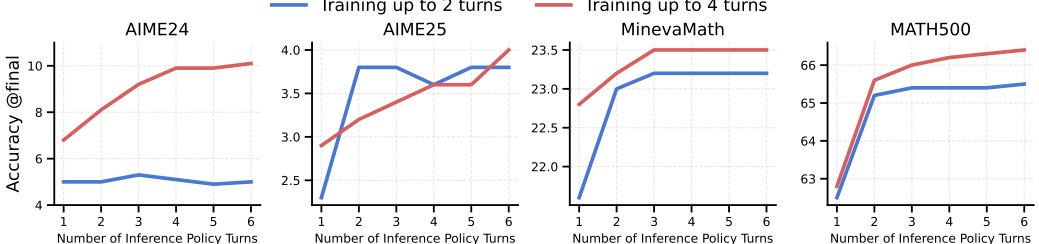

Figure 6: **Scaling Turns of PAG on Qwen2.5-1.5B-Instruct.**

over 40% for the 7B model. On RewardBench, the PAG-enhanced models not only outperform their respective baselines by large margins but also surpass several strong large-scale models, including Gemini (Team et al., 2023), Llama-3.1 (Grattafiori et al., 2024), and GPT-4 (Achiam et al., 2023). These findings highlight the effectiveness of the PAG framework in enhancing model self-verification, even when trained solely on policy data.

**PAG self-verify BoN outperforms majority voting.** We compare the performance of majority voting and self-verify BoN using the Qwen2.5-1.5B/7B-Instruct models on AIME24/25. As illustrated in Figure 5, PAG self-verify BoN consistently outperforms majority voting. This result is particularly noteworthy given that prior work (Huang et al., 2023) has demonstrated that self-verification typically underperforms compared to majority voting. Our findings indicate that this limitation no longer holds under the PAG framework, providing compelling evidence that the PAG training paradigm substantially enhances the model's self-verification capabilities.

### 4.3    SCALING TRAINING AND INFERENCE TURNS OF PAG

To better understand the scalability of our approach, we investigate two critical factors that influence model performance: the number of training policy turns and the number of inference policy turns. We extend the policy turns during inference to 6 and scale the training policy turns up to 4. Figure 6 presents the results of this analysis on the model trained using PAG with Qwen2.5-1.5B-Instruct.

Our findings indicate that while more inference turns generally lead to higher accuracy, the improvement is substantially greater when the model is trained for more turns. As illustrated in Figure 6, training with up to four turns yields significant gains as inference turns increase, whereas training with only two turns results in marginal improvements. This contrast is starkest on AIME24, where the four-turn model achieves 10.1% accuracy after six inference turns, compared to just 5.1% for the two-turn model.

### 4.4    ABLATION STUDIES

We conduct ablation studies on several key components of our method: (1) The selective revision mechanism—the core design principle of PAG. (2) Turn-independent optimization. (3) RoleAdvNorm and reward shaping techniques within our multi-turn RL algorithm.

**Selective revision is crucial.** We find that revising the initial solution attempt only when self-verification judges the previous answer as wrong is crucial for effective self-correction. We conduct ablation studies with two variants: (1) *PAG w/o revision*, which generates only a single response and a single verifier output (*i.e.*, no revision), and (2) *PAG w/ always revision*, which always revises the

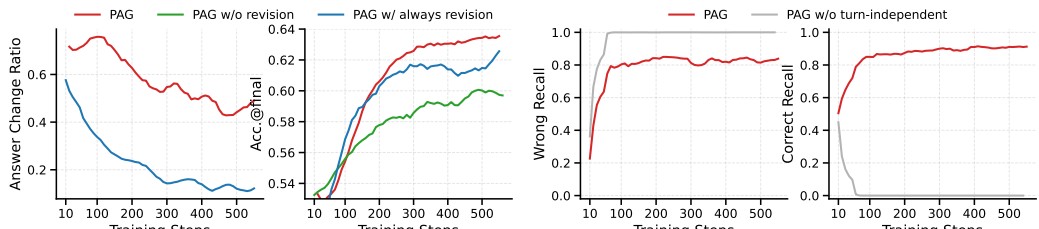

Figure 7: **Ablations of selective revision (Left) and turn-independent optimization (Right).** Selective revision and turn-independent optimization are crucial for effective self-correction.

Table 5: **Ablations of RoleAdvNorm and reward shaping.** Combining RoleAdvNorm and reward shaping yields the best self-correction performance, while using either alone brings no improvement.

| RoleAdvNorm | Reward Shaping | Acc.@t1 | Acc.@final |
|:---:|:---:|:---:|:---:|
| | | 62.0% | 64.7% |
| | ✓ | 61.5% | 64.4% |
| ✓ | | 61.8% | 64.7% |
| ✓ | ✓ | **62.5%** | **65.6%** |

initial attempt regardless of the self-verification outcome. As shown in the left of Figure 7, PAG significantly outperforms both *PAG w/o revision* and *PAG w/ always revision*. The superiority of *PAG* over *PAG w/o revision* is intuitive, as the latter lacks any self-correction mechanism. *PAG w/ always revision* essentially degenerates into Direct MultiTurn, and we observe that its answer change ratio decreases rapidly, which fundamentally constrains its self-correction capability.

**Turn-independent optimization prevents verifier collapse.** We compare our turn-independent optimization approach with a variant that propagates advantages across turns (*i.e.*, "w/o turn-independent optimization"). As shown in the right of Figure 7, *w/o turn-independent optimization* causes the generative verifier to collapse: the model always outputs "The answer is wrong", with correct recall dropping to zero and wrong recall rising to one. The core issue is that the verifier's reward signal can be contaminated by subsequent policy turns. When advantages are propagated across turns, the verifier can be rewarded for an incorrect judgment if it happens to precede a high-reward subsequent policy attempt. This leads to reward hacking, where the model learns to always trigger additional turns, regardless of the actual correctness of the answer. Turn-independent optimization decouples verifier and policy training, preventing such interference and ensuring stable verifier learning.

**The combination of RoleAdvNorm and reward shaping provides improvements.** We conduct ablation studies to examine the effects of reward shaping and task normalization (RoleAdvNorm) on Qwen2.5-1.5B-Instruct, evaluating on MATH500. As shown in Table 5, PAG achieves the best results when both reward shaping and RoleAdvNorm are applied together. In contrast, applying either reward shaping or RoleAdvNorm alone does not yield improvements. This can be attributed to the following: reward shaping is designed to enhance the model's performance in the second policy turn. However, because a verifier step is interleaved between the two policy turns, the verifier's advantage is also included in the advantage normalization, which may diminish the effect of reward shaping. When RoleAdvNorm is introduced, the verifier's advantage is excluded from the normalization process across the two policy turns, thereby allowing reward shaping to take full effect and improving overall performance.

## 4.5 PAG RESULTS ON LOGIC REASONING TASKS

Table 6: **Knights and Knaves Performance (%).**

| Method | Acc.@t1 | Acc.@final | $\Delta$ | $\Delta^{i \to c}$ | $\Delta^{c \to i}$ |
|:---|:---:|:---:|:---:|:---:|:---:|
| Qwen2.5-7B-Instruct | 13.8% | 14.3% | 0.5% | 1.0% | 0.5% |
| Direct MultiTurn | 65.1% | 66.1% | 1.3% | 1.3% | 0.3% |
| SCoRe | 62.4% | 65.0% | 2.8% | 2.8% | 0.2% |
| PAG | 79.9% | **82.1%** | 2.2% | 2.2% | 0.0% |

Table 7: **Countdown Performance (%).**

| Method | Acc.@t1 | Acc.@final | $\Delta$ | $\Delta^{i\to c}$ | $\Delta^{c\to i}$ |
|---|---|---|---|---|---|
| Qwen2.5-7B-Instruct | 0.0% | 0.8% | 0.8% | 0.8% | 0.0% |
| Direct MultiTurn | 64.1% | 69.2% | 5.1% | 6.3% | 1.2% |
| SCoRe | 67.6% | 70.8% | 3.2% | 4.6% | 1.4% |
| PAG | **74.3%** | **80.2%** | 5.9% | 6.5% | 0.6% |

We evaluated our approach on two logic reasoning benchmarks. The first is the **Knights and Knaves** logic puzzle (Xie et al., 2024a), where each character either always tells the truth or always lies. The task is to infer the truthfulness of each character. We used both training and test sets from the Hugging Face dataset `K-and-K/perturbed-knights-and-knaves`. Recent works (Xie et al., 2024b; Yang et al., 2024b) specifically use K&K puzzles as a deductive reasoning benchmark to assess LLM reasoning capabilities.

The second benchmark is the **Countdown language game**. It it a generalization of the 24-game, is widely adopted as a benchmark for arithmetic and search reasoning. Numerous recent studies (Gandhi et al., 2024; Feng et al., 2024; Katz et al., 2024; Yao et al., 2023) evaluate models' reasoning and search strategies on this family of problems. Given a target number and a set of numbers, the objective is to reach the target using arithmetic operations (addition, subtraction, multiplication, and division). We used the dataset construction script from `reasoning_gym` (Stojanovski et al., 2024) to generate a training set of 2,000 samples and a test set of 100 samples.

We trained all models using Qwen2.5-7B-Instruct and compared our PAG method against Direct MultiTurn and SCoRe. Results are presented below. Table values indicate the avg@8 performance per sample. PAG performs best in both Acc.@t1 and Acc.@final.

### 4.6 PAG RESULTS ON CODING REASONING TASKS

We conducted additional code generation experiments using Qwen2.5-7B-Instruct on the Hugging-Face dataset ganler/code-r1-12k. We evaluated on HumanEval/+, MBPP/+, and LiveCodeBench (v5), keeping all hyperparameters consistent with our math experiments. The results show that PAG remains competitive with SCoRe and even outperforms it on several metrics, despite being simpler: PAG trains end-to-end without the multi-stage procedure required by SCoRe.

Table 8: **Code Generation Performance (%).**

| Model | Metric | HumanEval | HumanEval+ | MBPP | MBPP+ | LiveCodeBench(v5) | Average |
|---|---|---|---|---|---|---|---|
| Qwen2.5-7B-Instruct | Acc.@t1 | 86.0 | 78.0 | 80.2 | 69.0 | 28.4 | 68.3 |
| | Acc.@final | 86.6 | 80.5 | 80.0 | 68.3 | 26.4 | 68.4 |
| SCoRe | Acc.@t1 | 84.1 | 79.8 | 86.2 | 72.5 | 33.0 | 71.1 |
| | Acc.@final | 86.3 | **81.1** | **86.8** | 73.5 | 33.9 | 72.3 |
| PAG | Acc.@t1 | 86.0 | 80.5 | 85.4 | 73.0 | 33.5 | 71.7 |
| | Acc.@final | **87.2** | **81.1** | 86.0 | **73.8** | **34.0** | **72.4** |

## 5 CONCLUSION

We introduce PAG (Policy as Generative Verifier), a novel framework efficiently training a single LLM to serve dual roles as generator and verifier for multi-turn self-correction. PAG streamlines training by circumventing complex warm-up stages and mitigating model collapse. Extensive experiments on mathematical reasoning benchmarks show that PAG outperforms existing methods in self-correction capabilities. Moreover, PAG exhibits strong self-verification: its self-verify best-of-N achieves superior performance over majority voting. One limitation of PAG is its reliance on external ground-truth verifiers to provide reward signals during training, which restricts its applicability to tasks where such supervision is unavailable.

## 6 REPRODUCIBILITY STATEMENT

We are committed to ensuring the reproducibility of our research. we provide the full source code for training in the supplementary material. All datasets used for training and evaluation are publicly available. Appendix C contains a comprehensive description of our experimental setup, including all hyperparameter settings (Table 13) and training configurations.

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

## A  THE USE OF LARGE LANGUAGE MODELS (LLMs)

We utilized a large language model to enhance the quality of this paper. Its applications included refining the manuscript for clarity, conciseness, and grammatical correctness.

## B  ADDITIONAL EXPERIMENTS

### B.1  SCALING TO QWEN2.5-32B-INSTRUCT

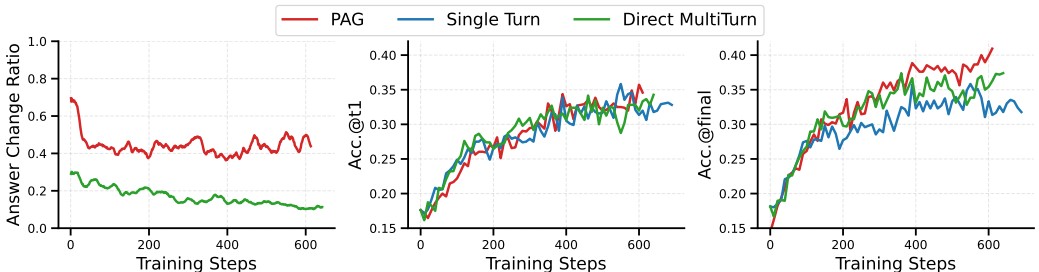

Figure 8: **Training dynamics of PAG on Qwen2.5-32B-Instruct. Left:** *Answer change ratio* quantifies model collapse as the proportion of responses where the second-turn answer differs from the first. Direct MultiTurn rapidly declines, indicating severe collapse. PAG's selective revision mechanism effectively prevents collapse and achieves higher Acc.@final; **Middle:** Acc.@t1; **Right:** Acc.@final.

Table 9: **Self-Correction Performance (%) of PAG on Qwen2.5-32B-Instruct.** PAG improves the revision efficiency.

| Method | Acc.@t1 | Acc.@final | $\Delta$(t1,final) | $\Delta^{i\to c}$ | $\Delta^{c\to i}$ | $\hat{\Delta}^{i\to c}$ | $\hat{\Delta}^{c\to i}$ |
|---|---|---|---|---|---|---|---|
| Base Model | 17.6 | 18.1 | 0.5 | 0.9 | 0.4 | 0.9 | 0.4 |
| SingleTurn | **35.8** | 35.8 | 0 | 1.5 | 1.5 | 1.5 | 1.5 |
| Direct MultiTurn | 32.7 | 37.4 | 4.7 | 7.2 | 2.5 | 7.2 | 2.5 |
| PAG | 34.6 | **40.9** | **6.3** | 7.8 | 1.5 | 13.3 | 2.5 |

We extend PAG to a larger model, Qwen2.5-32B-Instruct, and evaluate its performance on AIME2024 with a maximum response length of 8k. In training, keeping all hyperparameters consistent with our Qwen2.5-7B-Instruct experiments. Figure 8 displays the training dynamics, showing that PAG mitigates model collapse and achieves superior final accuracy. Table 9 demonstrates improved revision efficiency and the highest final accuracy, outperforming Direct MultiTurn by 3.5% (37.4 → 40.9).

### B.2  COMPARISON WITH DIRECT MULTI-TURN TRAINING USING THREE TURNS

We note that PAG's two-turn process consists of two policy generations and one verifier generation, resulting in three outputs from the LLM. In contrast, baseline Direct Multi-Turn methods typically involve only two policy generations. To ensure a fair comparison, we train the Direct Multi-Turn baseline with three turns and compare it to PAG's two-turn setting. It is important to highlight that, due to PAG's selective revision mechanism, not every problem requires three outputs, whereas Direct Multi-Turn always generates three outputs for every problem, making PAG more token-efficient. As shown in Table 10, increasing the number of training turns for Direct Multi-Turn from two to three leads to a higher final accuracy. However, PAG still achieves superior final accuracy, demonstrating its effectiveness both in terms of token efficiency and self-correction performance. We do not compare with SCoRe (Kumar et al., 2024), as it is specifically designed for two-turn settings.

| Method | Qwen2.5-1.5B-Instruct | |
| --- | --- | --- |
| | Acc.@t1 | Acc.@final |
| **Direct Multi-Turn (2 Turns)** | 62.1% | 64.0% |
| **Direct Multi-Turn (3 Turns)** | 62.3% | 64.5% |
| **PAG(Ours)** | 62.5% | **65.6%** |

Table 10: **Comparison of PAG and Direct Multi-Turn (2 and 3 turns) on MATH500 using Qwen2.5-1.5B-Instruct.** Increasing the number of training turns in Direct Multi-Turn improves Acc.@final, but PAG still achieves the best performance.

### B.3    SEQUENTIAL SELF-CORRECTION SAMPLING VS PARALLEL SAMPLING

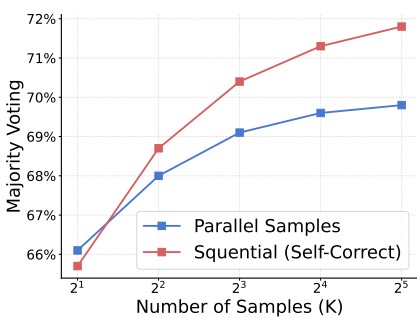

Figure 9: Sequential Sampling vs Parallel Sampling

We compare two paradigms for inference-time scaling: sequential self-correction sampling and parallel sampling. Sequential self-correction sampling generates $K$ solutions in parallel, then applies one round of self-correction to each solution, while parallel sampling directly samples $2K$ solutions in parallel. We employ majority voting as the aggregation metric for final answers, conducting experiments on Qwen2.5-1.5B-Instruct and evaluating performance on MATH500. As illustrated in Figure 9, sequential self-correction sampling demonstrates substantially superior computational efficiency compared to parallel sampling. Notably, sequential sampling with self-correction at $K = 8$ achieves performance superior to parallel sampling at $K = 32$, yielding a remarkable 4× improvement in computational efficiency. These results underscore the significant practical advantages of integrating sequential self-correction into the sampling framework, which aligns with recent findings in the literature (Snell et al., 2025; Kumar et al., 2024).

### B.4    ABLATION OF STANDALONE GENERATIVE VERIFIER: THE IMPORTANCE OF ONLINE TRAINING

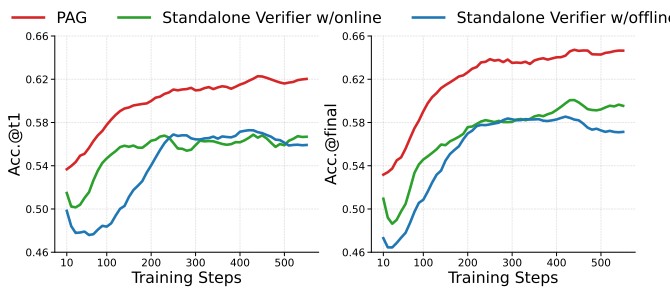

Figure 10: Policy Performance of Standalone Generative Verifier on MATH500

In PAG, the verifier is co-trained with the policy using on-policy data. This ablation study investigates the implications of training a generative verifier independently via reinforcement learning. We explore

| Model | Score |
|---|---|
| Qwen2.5-1.5B-Instruct | 23.9 |
| Qwen2.5-1.5B-Instruct + PAG | 78.5 |
| Qwen2.5-1.5B-Instruct + Standalone Verifier (online) | 78.1 |
| Qwen2.5-1.5B-Instruct + Standalone Verifier (offline) | 72.0 |

Table 11: Verifier Performance Standalone Generative Verifier on RewardBench mathprm.

two strategies for generating the verifier's training data: (1) **Standalone Verifier (offline)**: The Qwen1.5B-Instruct model initially generates four responses per MATH question. These responses, serving as prompts for the verifier's tasks, are labeled as correct or incorrect to constitute the verifier's training dataset. (2) **Standalone Verifier (online)**: Following each update to the policy model, new responses are generated for the same set of questions. These fresh responses then become the updated prompts for verifier training, and the verifier is trained on this dynamically generated data. This online process is analogous to two generation rounds in PAG but omits the revision step and does not leverage first-round outputs for policy optimization.

To evaluate policy performance, we employ the PAG framework. As illustrated in Figure 10, we report the policy results. While the standalone verifier does not surpass the full PAG framework, we find that training the verifier alone still enhances policy performance, as evidenced by improvements in both Acc.@t1 and Acc.@final. Notably, the Standalone Verifier w/online achieves a higher Acc.@final than its offline counterpart, suggesting that online training of the verifier is more effective by mitigating distribution shift.

To evaluate the verifier, we use the RewardBench prm dataset, with results presented in Table 10. We observe that the standalone verifier (offline) exhibits weaker verification capability compared to the standalone verifier (online), as evidenced by a 6.1-point lower score on RewardBench. This indicates that online training leads to a more effective verifier than offline training. Notably, the verification ability of the online-trained standalone verifier is already comparable to that of the full PAG verifier. PAG does not significantly improve the verifier's standalone capability; rather, it leverages the verifier's strength to enhance the overall efficiency and effectiveness of self-correction in the model.

## B.5 ABLATION OF USING GROUND-TRUTH VERIFIER INSTEAD OF SELF-VERIFICATION IN TRAINING

Table 12: **PAG with Ground-Truth Verifier.**

| | Acc.@t1 | Acc.@t2 | $\Delta$ | $\Delta^{i \to c}$ | $\Delta^{c \to i}$ |
|---|---|---|---|---|---|
| PAG | 62.2% | 64.5% | 2.3% | 3.6% | 1.3% |
| PAG(trained with ground-truth verifier) | 61.1% | 40.0% | -21.1% | 9.0% | 30.1% |

We experimented with using a **ground-truth verifier instead of self-verification** during training on math tasks to decide whether to revision. During evaluation, as we can not access ground-truth verifier, we prompt the model to always generate a second response—same to the Direct Multiturn and SCoRe approaches. The results are shown in the Table 12.

The results reveal that **PAG with ground-truth verification performs significantly worse on Acc.@t2 compared to Acc.@t1**. This is because a form of *training-time hacking* occurred. The ground-truth verifier is fixed, so during training, whenever a second answer is generated, it necessarily means the first answer was incorrect. As a result, the model learns to *always* revise its first answer. During evaluation, however, even when the first answer is correct, forcing a second attempt causes the model to revise it unnecessarily, often changing a correct answer into an incorrect one. This leads to a high **correct-to-incorrect ($\Delta^{c \to i}$) flip rate**.

These findings suggest that **using a *fixed* verifier during training is prone to similar hacking by the model**. In contrast, self-verification—where the verifier is itself trainable—avoids this issue, as the model must learn when revision is truly needed.

## B.6 ABLATION OF MULTI-TURN REWARD: THE IMPORTANCE OF PER-TURN REWARDS

In the standard multi-turn training paradigm of PAG, both policy and verifier outputs at each turn are assigned ground-truth rewards. In this section, we systematically investigate the effect of providing rewards only at specific turns, rather than at every turn. We consider two ablation variants: (1) **PAG w/ final reward only**: Only the final output—regardless of whether it is generated by the policy or the verifier—receives a reward. For this setting, the turn-level discount factor is set to 1. (2) **PAG w/o first policy reward**: The policy output at the first turn does not receive a reward; instead, the verifier output at the second turn is rewarded, and if a second policy output is generated, it also receives a reward. In this case, a turn discount of 1 is applied between the first policy output and the second verifier output, and training is truncated at the third turn.

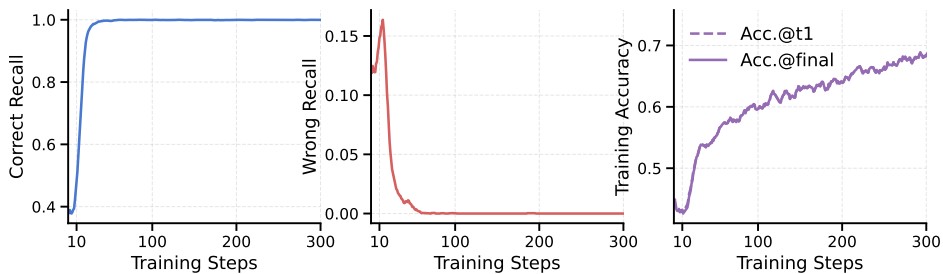

Figure 11: Ablation of PAG with final reward only.

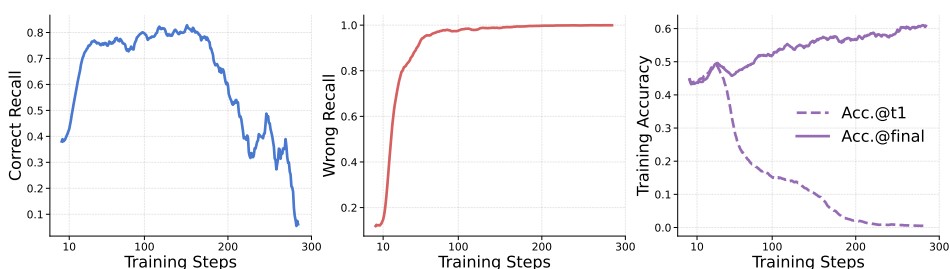

Figure 12: Ablation of PAG without first policy reward.

Figure 11 shows the results for PAG when only the final output is rewarded (PAG w/ final reward only). We observe that the verifier collapses to always output "The answer is correct," as indicated by Correct Recall converging to 1 and Wrong Recall dropping to 0. In this scenario, the model is incentivized to maximize Acc.@t1, since this directly increases the reward by encouraging the verifier to always confirm the answer as correct. This is further evidenced by the continuous increase of Acc.@t1 on the training set. As a result, Acc.@final becomes identical to Acc.@t1, since no revision occurs when the verifier always outputs "The answer is correct."

Figure 12 presents the results for PAG without the first policy reward (PAG w/o first policy reward). In this setting, the policy collapses to almost always output "The answer is wrong," with Wrong Recall rapidly decreasing to 0. This phenomenon occurs because, without a reward for the first policy output, the model is incentivized to produce arbitrary answers in the first round, prompting the verifier to label them as wrong and thus receive the verifier's reward. Subsequent revision then improves the answer, leading to a normal increase in Acc.@final.

In summary, our findings demonstrate that removing the reward at any turn leads to a collapse in either policy or verifier performance. Therefore, providing rewards at every turn is crucial for stable and effective training in PAG.

### B.7 ABLATION OF ROLEADVNORM AND REWARD SHAPING IN PAG

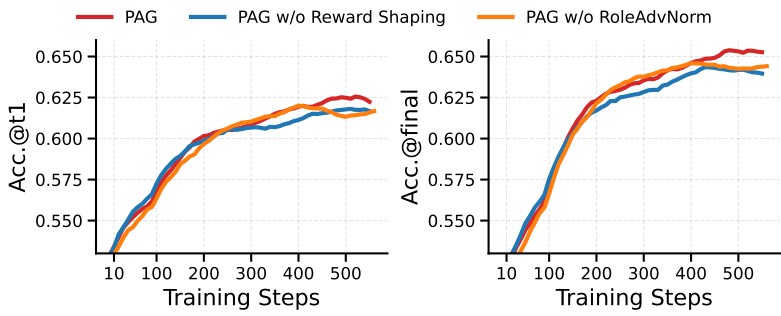

Figure 13: Ablation of RoleAdvNorm and reward shaping in PAG.

Table 5 shows that combining RoleAdvNorm with reward shaping produces the performance gains, whereas either component alone yields no improvement. To complement these results, we plot the training curves of PAG against PAG without RoleAdvNorm and PAG without reward shaping, further confirming that the joint use of both techniques provides a clear benefit. The underlying rationale is that reward shaping is intended to boost the model's performance on the second policy turn; however, because a verifier step is interleaved between the two policy turns, the verifier's advantage is also incorporated into the advantage normalization, potentially weakening the impact of reward shaping. By introducing RoleAdvNorm, the verifier's advantage is excluded from the normalization that spans the two policy turns, allowing reward shaping to exert its full effect and thereby enhancing overall performance.

### B.8 ABLATION OF KL IN PAG

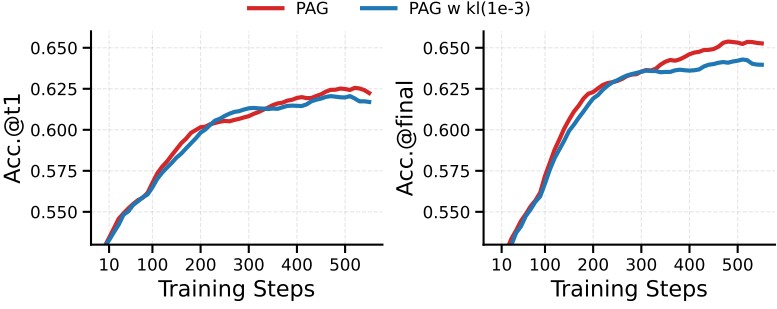

Figure 14: Ablation of KL in PAG.

Since our focus is on reasoning tasks with verifiable rewards, PAG drops KL regularization by default. Figure 14 compares PAG trained with a KL coefficient of 1e-3 against PAG without any KL term, and the latter reaches a clearly higher Acc.@final. This aligns with the common practice in RLVR, where removing KL penalties frees the model to explore more aggressively and boost the performance ceiling, whereas retaining KL penalties discourage that exploration.

## C IMPLEMENTATION DETAILS

Our implementation is based on the VeRL framework (Sheng et al., 2024), and we utilize vLLM 0.8.2 (Kwon et al., 2023) for our experiments. During reinforcement learning training, we do not

apply KL regularization. Table 13 summarizes the hyperparameters employed in our experiments. For reward shaping in both PAG and SCoRe, we use $\alpha = 1$; we have also experimented with $\alpha = 5$ and $\alpha = 10$, but observed no performance improvements. The maximum response length per turn in our experiments varies by model: 4096 tokens for Qwen2.5-7B-Instruct, 2048 tokens for Qwen2.5-1.5B-Instruct, and 2048 tokens for Llama-3-8B-Instruct. For the 1.5B model, we utilize 8×H800 GPUs with a training duration of less than 24 hours. For the 7B model, we employ 32xH800 GPUs with a training duration of less than 40 hours.

For evaluation, we generate 32 responses per prompt with temperature set to 0.6 , top-k to -1 and top-p to 0.95 (Sheng et al., 2024).

Table 13: **Hyperparameters used in our experiments During RL Training.**

| Hyperparameter | Value |
|---|---|
| Temperature | 1.0 |
| Top-p | 1.0 |
| Actor learning rate | $1 \times 10^{-6}$ |
| Critic learning rate | $2 \times 10^{-6}$ |
| Training prompt batch size | 512 |
| Responses per prompt | 4 |
| PPO epochs | 1 |
| PPO mini-batch size | 512 |
| PPO clip_high | 0.28 |
| PPO clip_low | 0.2 |
| Entropy loss coefficient | 0 |
| KL coefficient | 0 |

## D    EXAMPLES OF SELF-CORRECTION

All the cases are generate by the model trained on Qwen2.5-1.5B-Instruct.

### D.1    THE CASES IN FIGURE 2

---

**The Left in Figure 2** (SCoRe).

**System:**. Please reason step by step, and put your final answer within ☐.

**User:**. Alice and Bob are playing a game. Alice starts first. On Alice's turn, she flips a coin. If she gets a heads, she wins. If not, it becomes Bob's turn. On Bob's turn, he flips a coin. If he gets a tails, he wins. If not, it becomes Alice's turn. What is the probability that Alice wins the game?

**Assistant:**. First, let's define the probability that Alice wins on her first turn. Since Alice flips a coin and gets heads, she wins immediately with probability $\frac{1}{2}$.
Next, let's define the probability that Alice wins on Bob's turn. For Alice to win on Bob's turn, Alice must have lost on her previous turn, and then Bob must lose on his turn. The probability that Alice loses on her first turn is $\frac{1}{2}$, and the probability that Bob loses on his turn is $\frac{1}{2}$. Therefore, the probability that Alice wins on Bob's turn is:

$$\frac{1}{2} \times \frac{1}{2} = \frac{1}{4}$$

Now, let's define the probability that Alice wins on her second turn. For Alice to win on her second turn, both Alice and Bob must lose on their first turns, and then Alice must win on her second turn. The probability for this sequence is:

$$\left( \frac{1}{2} \times \frac{1}{2} \right) \times \frac{1}{2} = \frac{1}{8}$$

This pattern continues. The probability that Alice wins on her $n$-th turn is:

$$\left( \frac{1}{2} \right)^{2n+1}$$

The total probability that Alice wins the game is the sum of the probabilities of her winning on her first turn, her second turn, her third turn, and so on. This forms an infinite geometric series:

$$P(\text{Alice wins}) = \frac{1}{2} + \frac{1}{4} + \frac{1}{8} + \cdots$$

The sum of this infinite geometric series with the first term $a = \frac{1}{2}$ and common ratio $r = \frac{1}{2}$ is given by:

$$P(\text{Alice wins}) = \frac{\frac{1}{2}}{1 - \frac{1}{2}} = \frac{\frac{1}{2}}{\frac{1}{2}} = 1$$

However, we need to consider the probability that Alice wins on her first turn, on her second turn, etc. The correct probability that Alice wins the game is the sum of the probabilities of her winning on each of her turns, which is:

$$P(\text{Alice wins}) = \frac{1}{2} + \frac{1}{4} + \frac{1}{8} + \cdots$$

This is a geometric series with the first term $\frac{1}{2}$ and common ratio $\frac{1}{2}$. The sum of this series is:

$$P(\text{Alice wins}) = \frac{\frac{1}{2}}{1 - \frac{1}{2}} = \frac{\frac{1}{2}}{\frac{1}{2}} = 1$$

But since the series is infinite, we need to consider the probability that Alice wins on her first turn, which is $\frac{1}{2}$, and the probability that she wins on her second turn, which is $\frac{1}{4}$, and so on. The correct probability is actually the sum of the probabilities of her winning on each of her turns, which is:

$$P(\text{Alice wins}) = \frac{1}{2} + \frac{1}{4} + \frac{1}{8} + \cdots = \frac{\frac{1}{2}}{1 - \frac{1}{2}} = \frac{\frac{1}{2}}{\frac{1}{2}} = 1$$

But since the series is infinite, the correct probability is:

$$P(\text{Alice wins}) = \frac{1}{2}$$

Thus, the probability that Alice wins the game is:

$$\boxed{\dfrac{1}{3}}$$

**User:** There might be an error in the solution above because of lack of understanding of the question. Please correct the error, if any, and rewrite the solution.

**Assistant:** To determine the probability that Alice wins the game, we need to analyze the probability at each stage of the game.
1. **Alice's First Turn:** - The probability that Alice wins on her first turn is the probability that she gets a heads on her first coin flip. This probability is $\frac{1}{2}$.
2. **Bob's Turn:** - If Alice does not win on her first turn (probability $\frac{1}{2}$), then it is Bob's turn. The probability that Bob wins on his first turn is the probability that he gets a tails on his coin flip. This probability is $\frac{1}{2}$.
3. **Alice's Second Turn:** - If Bob does not win on his first turn (probability $\frac{1}{2}$), then it is Alice's second turn. The probability that Alice wins on her second turn is the probability that she gets a heads on her second coin flip and Bob gets a tails on his first coin flip. This probability is $\frac{1}{2} \times \frac{1}{2} = \frac{1}{4}$.
4. **General Case:** - The probability that Alice wins on her $n$-th turn is the probability that she does not win on her $(n-1)$-th turn and then wins on her $n$-th turn. This probability is $\left(\frac{1}{2}\right)^{2(n-1)} \times \frac{1}{2} = \left(\frac{1}{2}\right)^{2n-1}$.
The total probability that Alice wins the game is the sum of the probabilities of her winning on her first turn, her second turn, her third turn, and so on. This forms an infinite geometric series:

$$P(\text{Alice wins}) = \frac{1}{2} + \left(\frac{1}{2}\right)^3 + \left(\frac{1}{2}\right)^5 + \cdots$$

The first term $a$ of this series is $\frac{1}{2}$ and the common ratio $r$ is $\left(\frac{1}{2}\right)^2 = \frac{1}{4}$.
The sum $S$ of an infinite geometric series $a + ar + ar^2 + ar^3 + \cdots$ is given by $S = \frac{a}{1-r}$.
Substituting $a = \frac{1}{2}$ and $r = \frac{1}{4}$:

$$P(\text{Alice wins}) = \frac{\frac{1}{2}}{1 - \frac{1}{4}} = \frac{\frac{1}{2}}{\frac{3}{4}} = \frac{1}{2} \times \frac{4}{3} = \frac{2}{3}$$

Therefore, the probability that Alice wins the game is $\boxed{\dfrac{1}{3}}$.

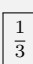

---

**The Middle in Figure 2** (PAG).

**System:** Please reason step by step, and put your final answer within ☐.

**User:** Alice and Bob are playing a game. Alice starts first. On Alice's turn, she flips a coin. If she gets a heads, she wins. If not, it becomes Bob's turn. On Bob's turn, he flips a coin. If he gets a tails, he wins. If not, it becomes Alice's turn. What is the probability that Alice wins the game?

**Assistant:** To determine the probability that Alice wins the game, we need to consider the sequence of turns and the probabilities associated with each player's turn.
First, let's define the probabilities:
The probability that Alice wins on her first turn is the probability that she gets a heads on her coin flip, which is $\frac{1}{2}$.

The probability that Bob wins on his first turn is the probability that he gets tails on his coin flip, which is $\frac{1}{2}$. If Bob gets tails, the game essentially resets with Alice's turn again.

Now, let's denote the probability that Alice wins the game as $P$.

If Alice doesn't win on her first turn (which happens with probability $\frac{1}{2}$), then it is Bob's turn. The probability that Bob doesn't win on his turn (which happens with probability $\frac{1}{2}$) is $\frac{1}{2}$. If Bob doesn't win, the game returns to the starting point, and the probability that Alice wins from this point onward is still $P$.

Thus, we can set up the following equation for the probability that Alice wins the game:

$$P = \frac{1}{2} \cdot 0 + \frac{1}{2} \cdot \left( \frac{1}{2} \cdot 0 + \frac{1}{2} \cdot P \right)$$

Simplifying the equation:

$$P = \frac{1}{2} \cdot \left( \frac{1}{2} P \right)$$

$$P = \frac{1}{4} P$$

To isolate $P$, we subtract $\frac{1}{4}P$ from both sides:

$$P - \frac{1}{4} P = 0$$

$$\frac{3}{4} P = 0$$

$$P = \frac{1}{3}$$

Therefore, the probability that Alice wins the game is $\boxed{\frac{1}{3}}$.

**User:** Check the math solution step-by-step. If you find a mistake: state the wrong step, explain why it's wrong, and end your response with 'The answer is wrong'. If all steps are correct, end your response with 'The answer is correct'.

**Assistant:** Let's break down the probability of Alice winning step by step.

The probability that Alice wins on her first turn is $\frac{1}{2}$.

If Alice does not win on her first turn, the game essentially resets to the same initial situation. The probability that Alice does not win on her first turn is $\frac{1}{2}$. If she doesn't win on her first turn, it becomes Bob's turn. The probability that Bob doesn't win on his turn is $\frac{1}{2}$, and if Bob doesn't win, the game returns to Alice's turn. The probability that Alice then wins is the same as the probability that she wins the game starting from the beginning, which we'll denote as $P$.

The probability that Alice wins can be expressed as the sum of probabilities of all scenarios that lead to Alice winning. The probability that Alice wins on her first turn is $\frac{1}{2}$.

The probability that Alice doesn't win on her first turn but then wins on her second turn is the probability that Alice gets tails on her first turn (which is $\frac{1}{2}$), then Bob gets tails (which is $\frac{1}{2}$), and then Alice wins. The probability of this sequence is $\frac{1}{2} \times \frac{1}{2} \times \frac{1}{2} = \frac{1}{8}$.

The probability that Alice wins on her third turn is more complex and involves the probability that the sequence of turns leading to Alice winning after two resets. The probability of Alice winning on the third turn is $\left( \frac{1}{2} \right)^3 \times \frac{1}{2} = \frac{1}{8}$.

We can generalize this as an infinite geometric series where the probability that Alice wins on her first turn is $\frac{1}{2}$, and the probability of her winning on subsequent turns is $\left( \frac{1}{2} \right)^n \times \frac{1}{2}$ for each n.

The total probability $P$ that Alice wins is the sum of an infinite geometric series:

$$P = \frac{1}{2} + \left( \frac{1}{2} \times \frac{1}{2} \right) + \left( \frac{1}{2} \times \left( \frac{1}{2} \right)^2 \right) + \cdots$$

This series can be summed as:

$$P = \frac{\frac{1}{2}}{1 - \frac{1}{4}} = \frac{\frac{1}{2}}{\frac{3}{4}} = \frac{1}{2} \times \frac{4}{3} = \frac{2}{3}$$

However, the simpler way to look at it is by breaking it into the two mutually exclusive scenarios: 1. Alice wins on her first turn: $\frac{1}{2}$ 2. If Alice doesn't win on her first turn, the game resets to the same initial condition with the same probability distribution. The probability that Alice wins from this new starting point is still $P$.

Thus, we can set up the equation as:

$$P = \frac{1}{2} + \frac{1}{2} \times \frac{1}{2} \times P$$

$$P = \frac{1}{2} + \frac{1}{4} P$$

Rearranging to solve for $P$:

$$P - \frac{1}{4} P = \frac{1}{2}$$

$$\frac{3}{4} P = \frac{1}{2}$$

$$P = \frac{1}{2} \times \frac{4}{3} = \frac{2}{3}$$

So, the probability that Alice wins the game is $\frac{2}{3}$.

The answer is wrong.

**User:** You indicated that your previous answer was wrong. Please provide the correct solution to the math problem.

**Assistant:** Let's break down the problem step by step to find the probability that Alice wins the game.

Alice wins on her first turn if she gets heads on her coin flip. The probability of Alice winning on her first turn is $\frac{1}{2}$.

If Alice does not win on her first turn, then it's Bob's turn. Bob wins on his first turn if he gets tails on his coin flip. The probability of Bob winning on his first turn is $\frac{1}{2}$.

If Bob doesn't win on his first turn (which happens with probability $\frac{1}{2}$), then it returns to Alice's turn. The game essentially resets with the same probability distribution. Therefore, the probability that Alice wins from this point onward is the same as the original probability $P$ that Alice wins the game.

We can set up the equation to represent the probability that Alice wins the game:

$$P = \frac{1}{2} + \frac{1}{2} \cdot \frac{1}{2} P$$

Here, $\frac{1}{2}$ is the probability that Alice wins on her first turn, and $\frac{1}{2} \cdot \frac{1}{2} P$ is the probability that the game resets and Alice still has the same probability $P$ to win from this point.

Simplifying the equation:

$$P = \frac{1}{2} + \frac{1}{4} P$$

To isolate $P$, we subtract $\frac{1}{4} P$ from both sides:

$$P - \frac{1}{4} P = \frac{1}{2}$$

$$\frac{3}{4} P = \frac{1}{2}$$

Now, solve for $P$ by dividing both sides by $\frac{3}{4}$:

$$P = \frac{1}{2} \cdot \frac{4}{3} = \frac{2}{3}$$

Therefore, the probability that Alice wins the game is $\boxed{\frac{2}{3}}$.

---

**The Right in Figure 2** (PAG).

**System:** Please reason step by step, and put your final answer within ☐.

**User:** For how many two-digit primes is the sum of the digits equal to 8?

**Assistant:** To determine how many two-digit prime numbers have digits that sum to 8, we first need to identify all two-digit numbers whose digits sum to 8. A two-digit number can be represented as $10a + b$ where $a$ and $b$ are its digits, and $a$ ranges from 1 to 9, while $b$ ranges from 0 to 9. We need $a + b = 8$.

Let's find all possible values of $a$ and $b$:
- If $a = 1$, then $b = 7$, giving the number 17.
- If $a = 2$, then $b = 6$, giving the number 26.
- If $a = 3$, then $b = 5$, giving the number 35.
- If $a = 4$, then $b = 4$, giving the number 44.
- If $a = 5$, then $b = 3$, giving the number 53.
- If $a = 6$, then $b = 2$, giving the number 62.
- If $a = 7$, then $b = 1$, giving the number 71.
- If $a = 8$, then $b = 0$, giving the number 80, but 80 is not a two-digit number.

So the two-digit numbers whose digits sum to 8 are: 17, 26, 35, 44, 53, 62, 71, and 80. However, we only need to consider the two-digit numbers, so we exclude 80. Thus, the valid two-digit numbers are: 17, 26, 35, 44, 53, 62, and 71.

Next, we need to check which of these numbers are prime:
- 17 is a prime number.
- 26 is not a prime number (divisible by 2 and 13).
- 35 is not a prime number (divisible by 5 and 7).
- 44 is not a prime number (divisible by 2 and 11).
- 53 is a prime number.
- 62 is not a prime number (divisible by 2 and 31).
- 71 is a prime number.

Thus, the prime numbers among the list are 17, 53, and 71. Therefore, the two-digit prime numbers whose digits sum to 8 are 17, 53, and 71.

Counting these, we find there are 3 such prime numbers. Therefore, the number of two-digit primes whose digits sum to 8 is $\boxed{3}$.

**User:** Check the math solution step-by-step. If you find a mistake: state the wrong step, explain why it's wrong, and end your response with 'The answer is wrong'. If all steps are correct, end your response with 'The answer is correct'.

**Assistant:** Let's first identify all two-digit numbers whose digits sum to 8. We need to find all numbers of the form $10a + b$ where $a$ and $b$ are digits (*i.e.*, $a$ ranges from 1 to 9 and $b$ ranges from 0 to 9), and $a + b = 8$.

We can list these numbers as follows: - If $a = 1$, then $b = 7$ giving the number 17.
- If $a = 2$, then $b = 6$ giving the number 26.
- If $a = 3$, then $b = 5$ giving the number 35.
- If $a = 4$, then $b = 4$ giving the number 44.
- If $a = 5$, then $b = 3$ giving the number 53.
- If $a = 6$, then $b = 2$ giving the number 62.
- If $a = 7$, then $b = 1$ giving the number 71.
- If $a = 8$, then $b = 0$ giving the number 80, but 80 is not a two-digit number.

So the two-digit numbers whose digits sum to 8 are: 17, 26, 35, 44, 53, 62, and 71. Now, we need to check which of these numbers are prime:
- 17 is a prime number.
- 26 is not a prime number (divisible by 2 and 13).
- 35 is not a prime number (divisible by 5 and 7).

## D.2 ADDITIONAL CASES OF PAG

### MATH EXAMPLE 1

**System:**. Please reason step by step, and put your final answer within □.

**User:**. A regular hexagon can be divided into six equilateral triangles. If the perimeter of one of the triangles is 21 inches, what is the perimeter, in inches, of the regular hexagon?

**Assistant:**. To determine the perimeter of the regular hexagon, we start by analyzing the properties of the equilateral triangles that make up the hexagon. Since the hexagon is divided into six equilateral triangles, each of these triangles has a side length equal to the side length of the hexagon.
We know that the perimeter of one of these equilateral triangles is 21 inches. Since an equilateral triangle has three equal sides, the side length of each equilateral triangle is:

$$\frac{21 \text{ inches}}{3} = 7 \text{ inches}$$

Each side of the equilateral triangles is therefore 7 inches. Since the hexagon is composed of six such equilateral triangles, the perimeter of the hexagon is six times the side length of one triangle:

$$6 \times 7 \text{ inches} = 42 \text{ inches}$$

Thus, the perimeter of the regular hexagon is $\boxed{42}$ inches.

**User:** Check the math solution step-by-step. If you find a mistake: state the wrong step, explain why it's wrong, and end your response with 'The answer is wrong'. If all steps are correct, end your response with 'The answer is correct'.

**Assistant:** Let's go through the problem step by step to find the perimeter of the regular hexagon.
First, we know that a regular hexagon can be divided into six equilateral triangles. If the perimeter of one of these equilateral triangles is 21 inches, we need to determine the side length of one of these equilateral triangles.
Since each equilateral triangle has three sides of equal length, the side length $s$ of one of these triangles is:

$$s = \frac{21 \text{ inches}}{3} = 7 \text{ inches}$$

Now, since a regular hexagon is composed of six such equilateral triangles, the perimeter of the hexagon is six times the side length of one equilateral triangle. So the perimeter $P$ of the hexagon is:

$$P = 6 \times s = 6 \times 7 = 42 \text{ inches}$$

Thus, the perimeter of the regular hexagon is 42 inches.
The answer is correct.

### MATH EXAMPLE 2

**System:** Please reason step by step, and put your final answer within □.

**User:** What is the remainder when $129^{34} + 96^{38}$ is divided by 11?

**Assistant:** To find the remainder when $129^{34} + 96^{38}$ is divided by 11, we start by simplifying the numbers modulo 11. First, we reduce 129 mod 11:

$$129 \div 11 = 11 \text{ remainder } 8 \Rightarrow 129 \equiv 8 \pmod{11}$$

Thus, $129^{34} \equiv 8^{34} \pmod{11}$.
Next, we reduce 96 mod 11:

$$96 \div 11 = 8 \text{ remainder } 8 \Rightarrow 96 \equiv 8 \pmod{11}$$

Thus, $96^{38} \equiv 8^{38} \pmod{11}$.
Now we need to find the remainders of $8^{34}$ mod 11 and $8^{38}$ mod 11.
By Fermat's Little Theorem, since 11 is a prime number, we know that for any integer $a$ such that $a$ is not divisible by 11, $a^{10} \equiv 1 \pmod{11}$. Thus, we can reduce the exponents modulo 10 because the exponent modulo 10 will give the same result for the remainder when divided by 11.
First, let's simplify $8^{34}$ mod 11:

$$34 \div 10 = 3 \text{ remainder } 4 \Rightarrow 34 \equiv 4 \pmod{10}$$

So, $8^{34} \equiv 8^4 \pmod{11}$.
Now, let's calculate $8^4$ mod 11:

$$8^2 = 64 \equiv 10 \pmod{11} \quad (\text{since } 64 - 5 \times 11 = 64 - 55 = 9)$$

$$8^4 = (8^2)^2 = 10^2 = 100 \equiv 100 - 9 \times 11 = 100 - 99 = 1 \pmod{11}$$

So, $8^4 \equiv 1 \pmod{11}$, therefore $8^{34} \equiv 1 \pmod{11}$.

Next, let's simplify $8^{38} \mod 11$:

$$38 \div 10 = 3 \text{ remainder } 8 \Rightarrow 38 \equiv 8 \pmod{10} \Rightarrow 38 \equiv 8 \pmod{10}$$

So, $8^{38} \equiv 8^8 \pmod{11}$.

Now, let's calculate $8^8 \mod 11$:

$$8^2 = 64 \equiv 9 \pmod{11} \quad (\text{since } 64 - 5 \times 11 = 64 - 55 = 9)$$

$$8^4 = (8^2)^2 = 9^2 = 81 \equiv 4 \pmod{11} \quad (\text{since } 81 - 7 \times 11 = 81 - 77 = 4)$$

$$8^8 = (8^4)^2 = 4^2 = 16 \equiv 5 \pmod{11} \quad (\text{since } 16 - 1 \times 11 = 16 - 11 = 5)$$

So, $8^8 \equiv 5 \pmod{11}$.

Now we add the two results:
$$8^{34} + 8^{38} \equiv 1 + 5 \equiv 6 \pmod{11}$$

Thus, the remainder when $129^{34} + 96^{38}$ is divided by 11 is $\boxed{6}$.

**User:** Check the math solution step-by-step. If you find a mistake: state the wrong step, explain why it's wrong, and end your response with 'The answer is wrong'. If all steps are correct, end your response with 'The answer is correct'.

**Assistant:** Let's break it down step by step:

First, we need to find the remainders of $129^{34}$ and $96^{38}$ modulo 11.

1. Simplify $129 \mod 11$:
$$129 \div 11 = 11 \text{ remainder } 8 \Rightarrow 129 \equiv 8 \mod 11$$

So, $129^{34} \equiv 8^{34} \mod 11$.

2. Simplify $96 \mod 11$:
$$96 \div 11 = 8 \text{ remainder } 8 \Rightarrow 96 \equiv 8 \mod 11$$

So, $96^{38} \equiv 8^{38} \mod 11$.

Now we need to find $8^{34} \mod 11$ and $8^{38} \mod 11$.

Using Fermat's Little Theorem which states that $a^{p-1} \equiv 1 \mod p$ for a prime $p$ and $a$ not divisible by $p$, we know that $a^{p-1} \equiv 1 \mod p$. Here $p = 11$, so $a^{10} \equiv 1 \mod 11$ for any $a$ not divisible by 11. So we can reduce the exponents modulo 10.

For $8^{34} \mod 11$:
$$34 \mod 10 = 4 \Rightarrow 8^4 \mod 11$$

Now we need to calculate $8^4 \mod 11$:

$$8^2 = 64 \equiv 64 - 5 \times 11 = 64 - 55 = 9 \mod 11$$

$$8^4 = (8^2)^2 = 9^2 = 81 \equiv 81 - 7 \times 11 = 81 - 77 = 4 \mod 11$$

So, $8^4 \equiv 4 \mod 11$.

For $8^{38} \mod 11$:
$$38 \mod 10 = 8 \Rightarrow 8^8 \mod 11$$

Now we need to calculate $8^8 \mod 11$:

$$8^2 = 64 \equiv 64 - 5 \times 11 = 64 - 55 = 9 \mod 11$$

$$8^4 = (8^2)^2 = 9^2 = 81 \equiv 81 - 7 \times 11 = 81 - 77 = 4 \mod 11$$

$$8^8 = (8^4)^2 = 4^2 = 16 \equiv 16 - 11 = 5 \mod 11$$

So, $8^8 \equiv 5 \mod 11$.

Now, adding these results:
$$8^{34} + 8^{38} \equiv 4 + 5 \equiv 9 \mod 11$$

Therefore, the remainder when $129^{34} + 96^{38}$ is divided by 11 is 9.

However, it seems there was an error in my previous calculation. Let's redo the calculations properly.

For $8^{34} \mod 11$:
$$8^{10} \equiv 1 \mod 11$$

So, $8^{34} = 8^{10 \cdot 3 + 4} = (8^{10})^3 \cdot 8^4 \equiv 1^3 \cdot 8^4 \equiv 8^4 \mod 11$. Now we calculate $8^4 \mod 11$:

$$8^2 = 64 \equiv 64 - 5 \cdot 11 = 64 - 55 = 9 \mod 11$$

$$8^4 = (8^2)^2 = 9^2 = 81 \equiv 81 - 7 \cdot 11 = 81 - 77 = 4 \mod 11$$

So, $8^4 \equiv 4 \mod 11$.

For $8^{38} \mod 11$:
$$8^{38} = 8^{10 \cdot 3 + 8} = (8^{10})^3 \cdot 8^8 \equiv 1^3 \cdot 8^8 \equiv 8^8 \mod 11$$

We already calculated $8^8 \equiv 5 \mod 11$.

Now adding them:
$$8^{34} + 8^{38} \equiv 4 + 5 \equiv 9 \mod 11$$

Thus, the remainder is 9.

The answer is wrong.

**User:** You indicated that your previous answer was wrong. Please provide the correct solution to the math problem.

**Assistant:** Let's break down the problem step by step.

We need to find the remainder when $129^{34} + 96^{38}$ is divided by 11.

First, we reduce the bases modulo 11:
$$129 \equiv 8 \pmod{11}$$

So, $129^{34} \equiv 8^{34} \pmod{11}$.

Next, we reduce 96 modulo 11:
$$96 \div 11 = 8 \text{ remainder } 8 \Rightarrow 96 \equiv 8 \pmod{11}$$

So, $96^{38} \equiv 8^{38} \pmod{11}$.

Now we need to find $8^{34} \mod 11$ and $8^{38} \mod 11$.

Using Fermat's Little Theorem, since 11 is a prime number, we know that $a^{p-1} \equiv 1 \pmod{p}$ for any integer $a$ not divisible by 11.

So for $a = 8$ and $p = 11$, we have:
$$8^{10} \equiv 1 \pmod{11}$$

Now, let's find the powers of 8 modulo 11:
$$8^2 = 64 \equiv 64 - 5 \times 11 = 64 - 55 = 9 \pmod{11}$$

So, $8^2 \equiv 9 \pmod{11}$.

Now we can find $8^{34} \mod 11$:
$$8^{34} = 8^{2 \cdot 17 + 0} = (8^2)^{17} \equiv 9^{17} \mod 11$$

Since $9 \equiv -2 \pmod{11}$, we have:
$$9^{17} \equiv (-2)^{17} \equiv -2^{17} \pmod{11}$$

We need to find $2^{17} \mod 11$. Using Fermat's Little Theorem, $2^{10} \equiv 1 \mod 11$, so:
$$2^{17} = 2^{10} \cdot 2^7 \equiv 1 \cdot 2^7 \equiv 2^7 \mod 11$$

Now, calculate $2^7 \mod 11$:
$$2^7 = 128 \equiv 128 - 11 \cdot 11 = 128 - 121 = 7 \pmod{11}$$

So, $2^7 \equiv 7 \pmod{11}$, thus:
$$9^{17} \equiv -7 \equiv -7 + 11 = 4 \pmod{11}$$

So, $8^{34} \equiv 4 \pmod{11}$.

Now, let's find $8^{38} \mod 11$:
$$8^{38} = 8^{2 \cdot 19 + 0} = (8^2)^{19} \equiv 9^{19} \mod 11$$

Since $9 \equiv -2 \pmod{11}$, we have:
$$9^{19} \equiv (-2)^{19} \equiv -2^{19} \pmod{11}$$

Now, we need to find $2^{19} \mod 11$:
$$2^{10} \equiv 1 \pmod{11}$$

So,
$$2^{19} = 2^{10} \cdot 2^9 \equiv 1 \cdot 2^9 \equiv 2^9 \mod 11$$

Now, calculate $2^9 \mod 11$:
$$2^9 = 512 \equiv 512 - 46 \cdot 11 = 512 - 506 = 6 \pmod{11}$$

So, $2^9 \equiv 6 \pmod{11}$, thus:
$$9^{19} \equiv -6 \equiv -6 + 11 = 5 \pmod{11}$$

So, $8^{38} \equiv 5 \pmod{11}$.

Now we add the two results:
$$8^{34} + 8^{38} \equiv 4 + 5 \equiv 9 \pmod{11}$$

Thus, the remainder when $129^{34} + 96^{38}$ is divided by 11 is $\boxed{9}$.

