# OpenReview forum: "PAG: Multi-Turn Reinforced LLM Self-Correction with Policy as Generative Verifier"
_ICLR.cc/2026/Conference — ICLR 2026 Conference Desk Rejected Submission_

### Official Review · Reviewer_A3KQ · 2025-10-23

**Soundness:** 2
**Presentation:** 3
**Contribution:** 1
**Rating:** 4
**Confidence:** 5

**Summary:**

This paper proposes PAG, a multi-turn reinforcement learning framework where the model alternates between generation and verification roles to perform self-correction. The key idea is to condition revision on the model’s own verification outcome by using a selective revision mechanism combined with turn-independent optimization and RoleAdvNoren, which stabilizes training and prevents the verifier from collapsing. The author evaluated PAG on multiple mathematical reasoning benchmarks including MATH500, MinervaMath, AIME (and some coding benchmarks in the Appendix), demonstrating gains over prior self-correction methods and single turn RL.

**Strengths:**

**[S1]** The paper is well written and the proposed method (PAG) is conceptually sound

**[S2]** The experiments are generally well executed across multiple datasets

**Weaknesses:**

**[W1: Unclear benefit over single-turn RL].** While multi-turn verification sounds intuitively useful, it remains unclear what concrete benefit it provides over strong single-turn RL methods (e.g., GRPO or PPO with a single verification step), which already achieve strong reasoning and self-verification performance [1]. Prior works have shown that single-turn RL-trained models can already conduct implicit self-checks or “rethink” their answers during inference [2]. Furthermore, the author should better clarify how its “single-turn” baseline is trained and ensure fair comparison with scaled GRPO/PPO baselines. Moreover, it would be valuable to show empirically whether multi-turn RL provides unique advantages beyond marginal performance gains.

**[W2: Missing baseline].** The paper lacks several important baselines. In particular, ReVISE [3], which is both efficient and effective (though based on offline RL), should be included for comparison against SCoRe [4]. Evaluating PAG against ReVISE would provide a fairer picture of its cost-effectiveness and practical advantage.

**[W3: Limited contribution of proposed components].** The proposed components (e.g., RoleAdvNorm and the reward shaping) appear to have limited impact. The ablation study suggests that their improvements are marginal and sometimes within noise level, as results under a single seed show very minor differences.

**[W4: Marginal overall gain while methodological complexity increases].**
The overall improvement appears too small relative to the complexity of the proposed framework (e.g., compared to SCoRe or direct multi-turn RL). Also, the combined effect of RoleAdvNorm and reward shaping produces less than a 1% improvement (up to 0.9% when used together). The benefit does not seem to justify the added algorithmic complexity. Typically, the improvement inthe  coding domain is even more marginal.

**[W5: Relatively done on small scale (i.e., larger reasoning model can have different trend)].** Since the experiments are mostly conducted on relatively small models, the observed improvements might be within noise, especially on benchmarks such as AIME. It would strengthen the paper to include experiments on mid-sized models (e.g., 32B) to confirm whether the observed trends persist at larger scales.

**[W6: Somewhat limited novelty].** The novelty of the work is somewhat limited. The concepts of self-verification and revision have already been explored in prior works [3, 4]. This paper mainly introduces engineering extensions to enable multi-turn training. The work would be more compelling if the authors could better highlight the unique advantages that arise specifically from the multi-turn formulation.

Reference\
[1] DeepSeekMath: Pushing the Limits of Mathematical Reasoning in Open Language Models\
[2] DeepSeek-V2: A Strong, Economical, and Efficient Mixture-of-Experts Language Model\
[3] ReVISE: Learning to Refine at Test-Time via Intrinsic Self-Verification, ICML 2025\
[4] Training language models to self-correct via reinforcement learning, ICLR 2025

**Questions:**

See the question above.

---

> ### Author Response · Authors · 2025-11-13
> **Authors' Rebuttal Part1**
>
> Thank you for the highly detailed review and constructive feedback about this work. Please find our response below.
>
> ### W1: Unclear benefit over single-turn RL
>
> As stated in the main text (L106–107), training PAG (Policy-as-Generative-Verifier) with a multi-turn RL paradigm **unifies policy generation and verifier capabilities in a single model**. The model demonstrates strong policy outputs (Section 4.1) and effective verifier behavior (Tables 3 & 4, Section 4.2). In contrast, single-turn RL primarily optimizes policy generation and lacks explicit verifier capacity. Notably, we achieve this without introducing any additional prompt data.
>
> Regarding [1], our understanding is that it evaluates self-consistency (i.e., majority voting) rather than self-verification performance; if we have misunderstood, please let us know.
>
> While [2] notes that single-turn RL-trained models can implicitly recheck or "rethink" answers at inference, our results demonstrate that multi-turn RL more effectively elicits such behavior, as supported by our experiments. **In Table 2, across six models, single-turn training yields Acc@t1 ≈ Acc@final, whereas PAG improves Acc@final by 1–2%**. Across four benchmarks with avg@32 sampling (MATH500: 500×32, MinervaMATH: 272×32, AIME24: 30×32, AIME25: 30×32; totaling 26,642 responses), this gain is consistent and robust. Additionally, **in Table 6, we evaluated Qwen2.5-32B-Instruct on AIME24 (avg@32): single-turn training yields Acc@t1 ≈ Acc@final, whereas PAG improves Acc@final by 5%.**
>
> ### W2: Missing baseline (ReVISE)
>
> Thank you for the reminder! We added a discussion of ReVISE in Related Work (L122–127, marked in blue). Both ReVISE and our approach use self-verification followed by revision to enhance self-correction. The key difference is that REVISE relies on multi-stage SFT/preference learning and custom data construction, whereas PAG achieves this through single-stage multi-turn RL without additional training stages or data warmup. As highlighted in our contribution (L103–104), this simplifies the training pipeline while maintaining strong self-correction.
>
> ### W3: Limited contribution of proposed components
>
> Our primary contribution is demonstrating that single-stage multi-turn RL suffices to elicit self-correction—without multi-stage schedules or warm-up data—thereby minimizing training intervention. RoleAdvNorm and reward shaping are supporting components rather than core contributions.
>
> On MATH500 (avg@32; 500×32 = 16,000 generations), these components(combined RoleAdvNorm and reward shaping) yield a 0.9% Acc.final improvement, which is statistically meaningful. To more clearly demonstrate the gains from RoleAdvNorm and reward shaping, **we further added a training curve comparison in the Appendix B.7.**(marked as blue). The results clearly show that combining RoleAdvNorm and reward shaping is beneficial.
>
> ### W4: Marginal overall gain while methodological complexity increases
>
> We argue that our method is simpler, not more complex. The verify-then-revise mechanism alleviates the model-collapse issue reported in multi-turn RL (e.g., SCoRe) without requiring multi-stage mitigation (as pointed out in  L128–138). We observe clear gains on mathematical (Table 2) and logical (Appendix B.2) reasoning compared to SCoRe , and comparable performance(Appendix B.3) on coding tasks. Given the simple and streamlined training procedure, this trade-off is reasonable—a point also noted positively by reviewer `p8QX`.
>
> RoleAdvNorm and reward shaping add minimal complexity—they require only almost one-line code changes (we have provided the code).

---

> ### Author Response · Authors · 2025-11-13
> **Authors' Rebuttal part2**
>
> ### W5: Include experiments on mid-sized models (e.g., 32B)
>
> **We have already included experiments on Qwen2.5-32B-Instruct in Appendix B.1**. On AIME24 (avg@32), Table 6 shows that PAG improves Acc.final by +3.5% over Direct MultiTurn (37.4 → 40.9) and +5.1% over SingleTurn (35.8 → 40.9), demonstrating effectiveness at mid-scale. Additionally, we note that the ReVISE experiments only include Llama-3.2-1B and Llama-3.1-8B—our experiments are already more comprehensive by comparison.
>
> ### W6: Somewhat limited novelty. This paper mainly introduces engineering extensions to enable multi-turn training.
>
> We respectfully disagree with the claim that "this paper mainly introduces engineering extensions to enable multi-turn training."
>
> First, ReVISE [3] and prior SFT-based methods require additional data construction for warm-up and involve multi-stage training. SCoRe [4] uses multi-turn RL to avoid extra data construction but still requires multi-stage training to prevent the model from degenerating in the second turn—simply copying the first-turn answer without refinement. **In contrast, PAG demonstrates that multi-stage training is unnecessary. We only need to perform single-stage multi-turn RL following a verifier-then-revise pipeline**. Our method fundamentally reduces training complexity.
>
> Second, PAG **integrates policy generation and verification capabilities into a single model.** Compared with SCoRe, it improves performance on mathematical and logical reasoning tasks while matching performance on coding tasks, and it simplifies the training complexity. Notably, we demonstrate that self-verification (self-verify Best-of-N) outperforms self-consistency (majority voting)—a finding that, to the best of our knowledge, has not been reported before [5-6].
>
> ### Reference
>
> [1] DeepSeekMath: Pushing the Limits of Mathematical Reasoning in Open Language Models.
>
> [2] DeepSeek-V2: A Strong, Economical, and Efficient Mixture-of-Experts Language Model.
>
> [3] ReVISE: Learning to Refine at Test-Time via Intrinsic Self-Verification. ICML 2025
>
> [4] Training language models to self-correct via reinforcement learning. ICLR 2025.
>
> [5] Large Language Models Cannot Self-Correct Reasoning Yet. ICLR2024.
>
> [6] Self-rewarding correction for mathematical reasoning.
>
> ---
>
> We thank you again for raising these important points. We believe the clarifications and additions strengthen the paper, and we welcome any further questions or suggestions.

---

### Official Review · Reviewer_p8QX · 2025-10-30

**Soundness:** 4
**Presentation:** 4
**Contribution:** 4
**Rating:** 8
**Confidence:** 5

**Summary:**

This paper proposes a method for training LLMs to self-correct by training the same model with two different roles: 1) a generative verifier that takes in a question and a solution and verifies the correctness of the solution, and 2) a solution policy that attempts to find the correct solution. The LLM is trained using multi-turn RL where the trajectories consist of an initial solution to the given question by the solution policy, then a verification by the generative verifier, followed by another solution attempt by the policy, and so on. Due to the fact that two roles are present in the same trajectory, and the fact that multiple turns are taken, the paper proposes several modifications to the standard PPO algorithm, consisting of 1) turn-independent optimization, 2) self-correction bonus reward, and 3) a role-based advantage normalization. The method proposed does not require any initial SFT or RL fine-tuning of the base model, which is an improvement over existing works on self-correction.

Extensive experiments on 3 base models (Qwen and LLama) and across several math, logic, and code generation benchmarks shows the method to improve final performance. Results also show that the training procedure also results in both a strong verifier, and improved self-correction capability, both an improvement over existing works.

**Strengths:**

1. The proposed method simplifies over existing works on LLM self-correction by unifying verification and solving within the same model. As mentioned in the summary this removes the need for any SFT or RL fine-tuning as a warm start to RL training, which is a significant improvement.

2. The changes made to RL training mentioned in the summary are quite reasonable and minor. They are also shown through ablations to be necessary for getting the proposed method to work well. They confirm findings from prior works (such as the self-correction bonus) while avoids adding more complexity.

3. Good breadth of models and benchmarks were studied in the paper, with consistent results. Baselines selected for study are appropriately strong.

4. Evidence of test-time improvement with increasing number of turns, even beyond the number of turns used during training.

**Weaknesses:**

1. The improvements from PAG are modest, especially in Tables 2 (math) and 9 (code generation) when compared do SCoRe. Although this still could be considered interesting as the method is simpler.

**Questions:**

Section 3.1, what does \hat{} mean in the notations? Seems unnecessary since there aren't same symbols with \hat{}?

Table 1, what's the evaluation benchmark? Where there only 2 turns in this setup (after initial solution)?

---

> ### Author Response · Authors · 2025-11-13
> **Authors' Rebuttal**
>
> Thank you for the highly detailed review and positive feedback about this work! Please find our response below.
>
> ### Q1: Section 3.1, what does \hat{} mean in the notations? Seems unnecessary since there aren't same symbols with \hat{}?
>
> Thank you for this careful observation. We originally introduced \hat{} to indicate that these symbols are estimates. However, as you note, in this context the decoration is indeed unnecessary, so we have removed it in the paper.
>
> ### Q2: Table 1, what's the evaluation benchmark? Where there only 2 turns in this setup (after initial solution)?
>
> The benchmark for Table 1 and Figure 4 is MATH500; thank you for pointing this out, and we have now explicitly stated it in the text. Unless otherwise specified, all experiments are two turns: an initial solution turn followed by a single revision turn.
>
> ---
> We thank you again for the positive feedback. we welcome any further questions or suggestions.

---

### Official Review · Reviewer_8hRm · 2025-10-30

**Soundness:** 3
**Presentation:** 3
**Contribution:** 3
**Rating:** 4
**Confidence:** 4

**Summary:**

The paper proposes policy as generative verifier (PAG), a multi-turn RL framework that trains a single LLM to alternate between two roles: (i) a policy that produces a solution, and (ii) a generative verifier that inspects that solution and decides whether a revision is needed ("verify-then-revise"). Unlike prior multi-turn methods that always produce a second attempt, PAG only revises when the model's own verification flags an error, aiming to avoid model collapse due to cosmetic edits or repeated answers while improving both reasoning and verification.

The method introduces three PPO-style adaptations: turn-independent optimization across turns, a bonus reward that encourages genuine improvements between attempts, and RoleAdvNorm, which normalizes advantages separately for the policy and verifier to prevent gradient interference. On mathematical reasoning (MATH500, MinervaMath, AIME24/25), PAG improves self-correction accuracy and shows strong verifier quality; notably, self-verify best-of-N (generate N candidates then let the unified verifier pick) outperforms majority voting.

**Strengths:**

1. This selective revision explicitly tackles collapse and yields better revision efficiency than always-revise baselines. Originality lies in unifying policy and verifier into one model with a *selective* verify-then-revise loop, plus stable multi-turn RL adaptations (turn-independent advantages + RoleAdvNorm + improvement bonus).
2. On Qwen2.5-7B, PAG reaches 82.3% Acc.@final on MATH500 and the best average 38.3%, outperforming Single-Turn, Direct Multi-Turn, and SCoRe; verifier performance on RewardBench also improves markedly (e.g., 86.6).
3. The self-verify BoN vs. majority voting result is practically useful.

**Weaknesses:**

1. Core results emphasize primarily on math; extensions to logic/coding are in the appendix but with much less emphasis.
2. The baselines are a bit "tricky", SCoRe is re-implemented (I know the code is not released), and for non-rl baselines such as self-consistency with majority voting at large N (policy-only BoN), the experiments are not under the same compute budget and decoding settings.
3. Appendix shows that dropping reward on either the first policy turn or final output collapses one role, I am concerned about robustness.

**Questions:**

1. Could you add main-paper results on at least one non-math domain (e.g., code repair or science QA)?
2. Could you include a small experiment comparing with/without KL?

---

> ### Author Response · Authors · 2025-11-18
> **Authors' Rebuttal**
>
> Thank you for the detailed review and constructive feedback about this work. Please find our response below.
>
> ### W1 & Q1: Core results emphasize primarily on math; extensions to logic/coding are in the appendix but with much less emphasis & Could you add main-paper results on at least one non-math domain.
>
> We have included experiments on logical and coding reasoning tasks, with results detailed in Appendix B.2 and B.3. **Due to the original page limit,** we placed these results in the appendix. On logical reasoning tasks (Appendix B.2), **we achieved a 17.1% improvement over SCoRe on Knights and Knaves (65.0%→82.1%)** and **a 9.8% improvement on CountDown (70.8%→80.2%).** On code reasoning tasks (Appendix B.3), PAG and SCoRe perform similarly. Meanwhile, PAG offers a simpler training procedure compared to SCoRe—it does not require multi-stage training. **Since the rebuttal period allows for 10 pages of main text, we have moved the logical and code reasoning results into the main paper (Section 4.5 & 4.6).**
>
> ### W2(part 1): The baselines are a bit "tricky", SCoRe is re-implemented.
>
> We have made every effort to ensure a faithful reproduction of the SCoRe baseline. Since SCoRe is a top-tier conference paper without publicly available code, we have re-implemented it and made our reproduction fully open-source in the supplementary materials, including reproduction scripts in `quick_start/qwen1p5b_SCoRe.sh`. Given that SCoRe has already garnered over 200+ citations and significant influence, we believe that providing this reproduction is also a meaningful contribution in itself.
>
> Additionally, we provide training curves for SCoRe in Figure 4, which closely align with Figure 6 in the original SCoRe paper:
>
> - Figure 6(a) in SCoRe shows that Stage 1 training causes the gap between Acc.@final and Acc.@t1 to widen progressively. We observe the same phenomenon in our Figure 4 when comparing Acc@t1 and Acc@final.
> - Figure 6(b) in SCoRe demonstrates that the answer change ratio in Stage 1 training is higher than in Direct MultiTurn, and that Stage 2 training does not reduce this ratio back to Direct MultiTurn levels. This behavior is also clearly visible in the left panel of our Figure 4.
>
> We welcome any specific feedback or corrections regarding the fairness of our baseline reproduction.
>
> ### W2(part 2): Majority voting vs self-verify BoN are not in the same compute budget.
>
> Thank you for this excellent question. Our main point is that **using the model as its own reward model has historically underperformed majority voting[1-2]**. However, after training with the PAG paradigm, the model becomes an effective reward model capable of selecting the best response—even from its own generations. **While self-verification inevitably requires additional tokens, previous work showed no benefit from this extra cost compared to majority voting. In contrast, PAG achieves better performance than majority voting despite the token overhead**, which we believe is a significant and interesting breakthrough.
>
> ### W3: Appendix shows that dropping reward on either the first policy turn or final output collapses one role, I am concerned about robustness.
>
> We respectfully disagree with this conern. **Per-turn rewards are a core design element of PAG (Section 3.1).** Figure 4 presents training curves for PAG on Qwen2.5B-Instruct across **three repeated runs; the shaded area shows metric variance, demonstrating consistent and stable training.** Figure 8 shows the training process on Qwen2.5-32B-Instruct, confirming that **PAG remains stable when scaled to a 32B model.**
>
> We include this experiment in Appendix B.8 to demonstrate that per-turn rewards are essential to PAG. **This ablation study is itself a key part of our method's contribution.**
>
> ### Q2: Could you include a small experiment comparing with/without KL?
>
> We appreciate this suggestion and have added an experiment comparing training with and without KL regularization in Appendix B.8 (marked in blue). Figure 14 compares PAG trained with a KL coefficient of 1e-3 against PAG without any KL term. The latter achieves a clearly higher Acc@final. This aligns with common practice in RLVR, where removing KL penalties allows the model to explore more aggressively and reach a higher performance ceiling, whereas retaining KL penalties tends to constrain exploration and limit potential gains.
>
> Referece:
>
> [1] Large Language Models Cannot Self-Correct Reasoning Yet.
>
> [2] Self-rewarding correction for mathematical reasoning.
>
> ---
> We would like to thank you again for raising important questions about this work, and we believe the additional clarifications and experiments have strengthened the paper. Please let us know if you have any additional concerns or questions.

---

### Official Review · Reviewer_doKC · 2025-10-31

**Soundness:** 4
**Presentation:** 4
**Contribution:** 2
**Rating:** 6
**Confidence:** 3

**Summary:**

The paper proposes PAG, a multi-turn reinforcement learning framework that enables an LLM to iteratively generate, verify, and revise its own reasoning outputs. By introducing a selective verify-then-revise mechanism, PAG avoids redundant second passes and stabilizes training compared to prior methods like SCoRe. The framework extends PPO with turn-independent optimization, a self-correction reward bonus, and RoleAdvNorm for separate policy and verifier updates. Experiments on various math reasoning benchmarks with Qwen-2.5 and Llama-3 show that PAG improves both reasoning accuracy and verifier quality, achieving superior test-time scaling through self-verifying Best-of-N sampling.

**Strengths:**

1. The paper is well written and clearly presented, making it easy to follow.

2. The model introduces three concrete and well-motivated components—turn-independent optimization, bonus reward, and
RoleAdvNorm—which are thoughtfully engineered to stabilize multi-turn RL training.

3. The empirical results are strong against baselines, especially in test-time scaling against majority voting.

4. The contributions are empirically well-demonstrated through detailed ablation studies and analysis.

5. The method shows strong empirical performance against baselines, particularly in test-time scaling, where it outperforms majority voting with self-verifying Best-of-N sampling.

**Weaknesses:**

1. The evaluation is limited to mathematical reasoning (Table 2), lacking experiments on logical or commonsense reasoning tasks that could better demonstrate generality.

2. The most significant concern is the absence of strong single-turn training baselines such as GRPO, DAPO, or DPO. To convincingly argue for the efficiency and effectiveness of training, comparisons with these methods are needed.

3. In Figure 5, the test-time scalability comparison between self-verify BoN and majority voting may be unfair, since self-verify incurs additional latency and token generation due to the verification step. A fair comparison should control for the same latency or total generation tokens.

**Questions:**

1. What is the training computational cost of PAG compared to strong single-turn RL methods GRPO or DAPO? For a fair comparison, it would be helpful to report training-time, FLOPs or total generation tokens used during both training and test-time scaling.

2. How does PAG behave when extended beyond mathematical reasoning? For example, would the verify–revise dynamic still hold on logical or commonsense reasoning tasks where correctness is less binary?

3. In Figure 5, could the authors provide a comparison under equal latency or token budget conditions to make the self-verify vs. majority voting comparison fairer?

---

> ### Author Response · Authors · 2025-11-18
> **Authors' Rebuttal**
>
> Thank you for the positive feedback about this work. Please find our response below.
>
> ### W1 & Q2: The evaluation is limited to mathematical reasoning, lacking experiments on logical or commonsense reasoning tasks that could better demonstrate generality.
>
> We have included the experiments on logical and coding reasoning tasks, with results detailed in Appendix B.2 and B.3. On logical reasoning tasks, we achieved a 17.1% improvement over SCoRe on Knights and Knaves (65.0%→82.1%) and a 9.8% improvement on CountDown (70.8%→80.2%). On code reasoning tasks, PAG and SCoRe perform similarly. Meanwhile, PAG offers a simpler training procedure compared to SCoRe—it does not require multi-stage training. **Since the rebuttal period allows for 10 pages of main text, we have moved the logical and code reasoning results into the main paper.**
>
> ### W2 & Q1: Compared to strong single-turn training baselines such as GRPO, DAPO, or DPO.
>
> We have included a single-turn baseline in Table 2, where the underlying RL algorithm is PPO. We chose PPO because it can be straightforwardly extended to multi-turn settings, which allows us to fairly compare different multi-turn training pipelines (Direct MultiTurn & SCoRe). We would like to emphasize that the PPO algorithm we employed is also a strong baseline (see VC-PPO[1] and VAPO[2]), and it surpasses GRPO/DAPO in performance.
>
> If we were to use GRPO as the single-turn baseline, we would need to adopt GRPO for multi-turn training as well to ensure a fair comparison. However, applying GRPO (or DAPO) in a multi-turn manner requires sampling a group of responses at each turn, which is highly inefficient. Thus, developing new multi-turn advantage normalization strategies for GRPO would be needed, but this falls beyond the scope of this paper.
>
> Regarding DPO, it is a preference-based learning method, whereas our focus is on RLVR (Reinforcement Learning from Verifiable Rewards) for reasoning tasks where a ground truth verifier is available. Notably, RLVR has become mainstream in 2025 [3-5].
>
> Below we compare token budget and computational overhead using Qwen 2.5-1.5B-Instruct as an example.
>
> **Training Comparison:** Training time for each method on Qwen 2.5-1.5B-Instruct (8×H800 GPUs), as reported in Table 2:
>
> |  | SingleTurn | DirectMultiTurn & SCoRe | PAG |
> | --- | --- | --- | --- |
> | Training Time (hours) | 10 | 18 | 22 |
>
> **Evaluation Comparison:** To ensure a fair token-budget comparison at evaluation time, we evaluated SingleTurn and DirectMultiTurn/SCoRe by running inference for 3 turns (matching PAG's two-turn inference) on MATH500:
>
> |  | Acc.final | Average Tokens Per Prompt |
> | --- | --- | --- |
> | Single-Turn | 62.5 | 1691 |
> | Direct-MultiTurn | 64.2 | 2665 |
> | SCoRe | 64.3 | 2433 |
> | PAG | 65.6 | 1883 |
>
> As shown above, **under similar evaluation token budgets, PAG consistently outperforms the baselines in Acc.@final** while remaining computationally efficient.
>
> Furthermore, in Appendix B.2 (formerly B.4; logical and coding reasoning results from the original B.2 and B.3 have been moved to the main text), we compare Direct MultiTurn trained and evaluated over three turns. The results show that **PAG still outperforms Direct MultiTurn even when the latter is trained  for three turns**. This demonstrates that PAG's verify-then-revise pipeline is more efficient than direct multi-turn training , while also avoiding the multi-stage training complexity required by SCoRe.
>
> ### W3 & Q3:  self-verify BoN and majority voting may be unfair, since self-verify incurs additional latency and token generation due to the verification step
>
> Our main point is that **using the model as its own reward model has historically underperformed majority voting[6-7]**. However, after training with the PAG paradigm, the model becomes an effective reward model capable of selecting the best response—even from its own generations. **While self-verification inevitably requires additional tokens, previous work showed no benefit from this extra cost compared to majority voting. In contrast, PAG achieves better performance than majority voting despite the token overhead**, which we believe is a significant and interesting breakthrough.
>
> Reference:
>
> [1] What’s Behind PPO’s Collapse in Long-CoT? Value Optimization Holds the Secret.
>
> [2] VAPO: Efficient and Reliable Reinforcement Learning for Advanced Reasoning Tasks.
>
> [3] DeepSeek-R1: Incentivizing Reasoning Capability in LLMs via Reinforcement Learning.
>
> [4] Kimi k1.5: Scaling Reinforcement Learning with LLMs.
>
> [5] Seed1.5-Thinking: Advancing Superb Reasoning Models with Reinforcement Learning.
>
> [6] Large Language Models Cannot Self-Correct Reasoning Yet.
>
> [7] Self-rewarding correction for mathematical reasoning.
>
> ---
> We would like to thank you again for raising important questions about this work, and we believe the additional clarifications have strengthened the paper. Please let us know if you have any additional concerns or questions.

---

### Note · Program_Chairs · 2026-01-17
**Submission Desk Rejected by Program Chairs**

The following references in this submission do not refer to real documents and/or have major errors in bibliographic information:

 Tian Xie, Xinyu Zhang, Yuesong Kong, Yike Wu, Zhi-Jie Wang, Kenji Kawaguchi, Ge-Li Philips, Yixin Zhu, and Song-Chun Zhu. Logic-RL: Unleashing LLM Reasoning with Rule-Based Reinforcement Learning, 2024ba